# Balancing Context Length and Mixing Times for Reinforcement Learning at Scale

**Matthew Riemer**[*]
IBM Research, Mila, Université de Montréal

**Khimya Khetarpal**
Mila

**Janarthanan Rajendran**[†]
Dalhousie University

**Sarath Chandar**
Mila, École Polytechnique de Montréal

## Abstract

Due to the recent remarkable advances in artificial intelligence, researchers have begun to consider challenging learning problems such as learning to generalize behavior from large offline datasets or learning online in non-Markovian environments. Meanwhile, recent advances in both of these areas have increasingly relied on conditioning policies on large context lengths. A natural question is if there is a limit to the performance benefits of increasing the context length if the computation needed is available. In this work, we establish a novel theoretical result that links the context length of a policy to the time needed to reliably evaluate its performance (i.e., its mixing time) in large scale partially observable reinforcement learning environments that exhibit latent sub-task structure. This analysis underscores a key tradeoff: when we extend the context length, our policy can more effectively model non-Markovian dependencies, but this comes at the cost of potentially slower policy evaluation and as a result slower downstream learning. Moreover, our empirical results highlight the relevance of this analysis when leveraging Transformer based neural networks. This perspective will become increasingly pertinent as the field scales towards larger and more realistic environments, opening up a number of potential future directions for improving the way we design learning agents.

## 1   Introduction

**Scaling the Context Length:** As the field of AI moves towards harder problems where agents must model higher order non-Markovian dependencies, it is only natural to consider conditioning models on larger context lengths of the interaction history. When considering only small context lengths or single observations, models become restricted to some lesser notion of the best possible performance in comparison to what is achievable with an infinite context length [1]. Meanwhile, there are two obvious downsides to increasing the context length an agent considers. The first downside is the increase in necessary computation. However, there is also ongoing research related to minimizing the needed quantity of computation [2; 3], increasing the parallelism of computation [4; 5; 6], and increasing the throughput of computation on modern hardware [7]. The second downside is the high dimensionality of the input representation we must process and the difficulty of learning and generalizing from this kind of input. But it is unclear how worried researchers and practitioners should be about this downside given the recent improvements in processing long sequences that came with attention models [8; 9] and particularly the Transformer architecture [10]. Additionally, there has been a trend towards training on internet-scale datasets [11; 12; 13] that far exceed the quantity

---

[*]Please direct correspondences to `mdriemer@us.ibm.com`.
[†]Work done during Postdoc at Mila, Université de Montréal.

38th Conference on Neural Information Processing Systems (NeurIPS 2024).

of any single human's experience in domains of interest, which significantly alleviates requirements for generalization. Our paper aims to add to this discourse by being the first to establish another downside: policies that are conditioned on longer context lengths take longer to reliably evaluate because they impose looser bounds on the mixing time. This may make it less safe to deploy these policies in the real-world because we are able to extrapolate less about their true behavior from our limited evaluations. Downstream learning may also be slower for the same reason as Monte Carlo algorithms can only perform reliable updates after being rolled out for the mixing time number of steps and the convergence time of TD algorithms also depends on the mixing time.[3]

**Addressing High Mixing Times:** Recent work has highlighted the challenges presented by high mixing times as the field moves towards large scale and continual reinforcement learning problems [14]. Unfortunately, as highlighted by Riemer et al. [14] there are not many known solutions that address this problem. As a result, it is important for researchers to consider how changes to the policy architecture and learning algorithms may have an impact on mixing behavior. The effect that the input variables sent to a policy could have on bounds on the mixing time was previously considered by the pioneering work of Kearns and Koller [15]. In Theorem 1 we present a tighter version of the bound presented in their paper. Moreover, our paper is the first to highlight the connection between this result and the context length in partially observable settings in Theorem 2. Our paper focuses solely on how design decisions related to the policy's architecture can impact the mixing time and is complementary to recent insights about how changes to the learning algorithm, such as incorporation of replay [16] or multi-level critics [17], can lead to faster learning when mixing times are high.

**Balancing Context Length and Mixing Times:** The core insight of this paper is that there is a tradeoff to consider between increasing the context length and increasing the mixing time. When we increase the context length sent to an agent, we increase its ability to model non-Markovian dependencies. However, this increase in context length may also cause an increase in the mixing time, which makes it take longer to reliably evaluate an agent's policy. Dong et al. [18] propose the solution of restricting the class of policies they optimize over such that the mixing times are bounded, which is also the perspective we embrace in this work. However, Dong et al. [18] restrict this class of policies by limiting the planning horizon and thus making optimization more myopic. In our work, we highlight a different mechanism for restricting the mixing times of the policies we optimize over by simply limiting the context length. To this end, **our key contributions** are as follows:

1. We present theoretical analysis with supporting toy examples shedding light on how the input sent to a policy can influence its mixing time in Section 2.

2. In Section 3 we formally establish the connection between the context length considered by a policy in partially observable environments and the mixing time. We also empirically verify the relevance of this theory to Transformer based agents in partially observable settings.

3. Finally, in Section 4 we highlight the relevance of this theory when building foundation models that imitate a diverse set of behavior policies for RL. Specifically, we show that Decision Transformers [19] must use much larger context lengths than the policies that generated their dataset, and as a result require more interaction for reliable evaluation.

To reproduce our experiments see https://github.com/mattriemer/ContextLengthMixing.

## 2 Understanding How the Policy's Input Impacts the Mixing Time

Before discussing the connection between mixing times and context length in the next section, we first provide the technical grounding for readers to understand how the problem structure and the policy combine to jointly affect bounds on the mixing time. We will begin by detailing the problem formulation and present a novel upper bound for the mixing time in this context in Theorem 1. We then highlight key implications with illustrative example problems.

### 2.1 Preliminaries: the Average Reward RL Setting and Definitions of the Mixing Time

**Problem Formulation:** We adopt the **average reward** setting to enable long-term analysis of agents. Please note that we only use this perspective to evaluate an agent's behavior and place no restriction

---

[3]See Riemer et al. [14] for a discussion of the difficulties presented for learning when mixing times are high.

on using discounting within the learning process (as we do in our experiments).[4] We consider a finite, discrete-time, infinite horizon Markov Decision Process (MDP) [20; 21]: $\mathcal{M} = \langle \mathcal{S}, \mathcal{A}, T, R \rangle$, where $\mathcal{S}$ is the set of states, $\mathcal{A}$ is the set of actions, $R : \mathcal{S} \times \mathcal{A} \to [0, R^{\max}]$ is the reward function, and $T : \mathcal{S} \times \mathcal{S} \times \mathcal{A} \to [0, 1]$ is the environment transition probability function. At each time step, an agent perceives a state $s \in \mathcal{S}$ and takes an action $a \in \mathcal{A}$ drawn from a policy $\pi : \mathcal{S} \times \mathcal{A} \to [0, 1]$. The agent then receives a scalar reward drawn from the function $R(s, a)$ and with probability $T(s'|s, a)$ enters next state $s'$. Markov chains may be periodic and have multiple recurrent classes, but optimality is difficult to define in such cases [22], making the following assumption necessary:

**Assumption 1** *All stationary policies are aperiodic and unichain, resulting in a Markov chain with a single recurrent class that is recurrent in the Markov chain of every policy.*[5]

Any RL problem may be modified such that Assumption 1 holds by adding an arbitrarily small positive constant $\epsilon$ to all transition probabilities in $T(s'|s, a)$ [23]. An important corollary is that the *steady-state distribution* $\mu^\pi$ induced by the policy $\pi$ is independent of the initial state:

**Corollary 1** *All policies $\pi$ induce a unique steady-state distribution $\mu^\pi(s) = \lim_{t \to \infty} P^\pi(s_t = s|s_0)$ that is independent of the initial state such that $\sum_{s \in \mathcal{S}} \mu^\pi(s) \sum_{a \in \mathcal{A}} \pi(a|s) T(s'|s, a) = \mu^\pi(s') \quad \forall s' \in \mathcal{S}.$*

Corollary 1 implies that long-term rewards and thus the average reward per step objective $\rho(\pi)$ can be defined independently of the current state [21]:

$$\rho(\pi) := \lim_{m \to \infty} \frac{1}{m} \sum_{t=1}^{m} \mathbb{E}_\pi \left[ R(s_t, a_t) \right] = \lim_{t \to \infty} \mathbb{E}_\pi \left[ R(s_t, a_t) \right]$$

$$= \sum_{s \in \mathcal{S}} \mu^\pi(s) \sum_{a \in \mathcal{A}} \pi(a|s) R(s, a) .$$

Computing the average reward (i.e., the reward rate) with the last expression is limited by the amount of time the Markov chain induced by the policy $T^\pi(s'|s) = \sum_{a \in \mathcal{A}} \pi(a|s) T(s'|s, a)$ needs to be run for before reaching the distribution $\mu^\pi(s)$. This amount of time is called the mixing time of the induced Markov chain. We denote $t_{\mathrm{mix}}^\pi(\epsilon)$ as the $\epsilon$-*mixing time* of the chain induced by $\pi$:

$$t_{\mathrm{mix}}^\pi(\epsilon) := \min \left\{ m \;\middle|\; \max_{s_0 \in \mathcal{S}} d_{\mathrm{TV}} \big( P^\pi(s_m = \cdot|s_0), \mu^\pi(\cdot) \big) \leq \epsilon \right\}$$

where $d_{\mathrm{TV}}$ is the total variation distance between the two distributions. The *conventional mixing time* is defined as $t_{\mathrm{mix}}^\pi \equiv t_{\mathrm{mix}}^\pi(1/4)$. This only gives insight about distributional mismatch with respect to the steady-state distribution, which led Kearns and Singh [24] to introduce the notion of a mismatch with respect to the reward rate. The $\epsilon$-*return mixing time* is a measure of the time it takes to formulate an accurate estimate of the true reward rate. Formally, if we denote the $m$-step average return starting from state $s_0$ as $\rho(\pi, s_0, m) := \mathbb{E}_\pi[\frac{1}{m} \sum_{t=1}^{m} r_t|s_0]$, then we can define the $\epsilon$-*return mixing time* as:

$$t_{\mathrm{ret}}^\pi(\epsilon) := \min \Big\{ m \;\Big|\; |\rho(\pi, s_0, m') - \rho(\pi)| \leq \epsilon,$$

$$\forall s_0 \in \mathcal{S} \text{ and } \forall m' \geq m \Big\}.$$

**An Interpretable Metric:** We will use this definition of the mixing time in our experiments as it directly measures the amount of time needed to evaluate a policy at a given threshold of precision $\epsilon$.

## 2.2 Mixing in MDPs with Multidimensional States

**Multidimensional State Space:** We will also assume that problems have a state spaces comprised of multiple dimensions. Formally, each state is separated into $n \geq 1$ variables i.e., $s = [s^1, ..., s^n]$ with $s^i \in \mathcal{S}^i$ for all variables $i \in \{1, ..., n\}$ such that $\mathcal{S} \subseteq \mathcal{S}^1 \times ... \times \mathcal{S}^n$. It is also important to note that each state variable can be modeled independently based on the previous state across all variables such that $T(s'|s, a) = \Pi_{i \in \{1, ..., n\}} T(s'^i|s, a)$. In Appendix A.1 we discuss why this setting is strictly more general than the typical discrete MDP setting and does not limit the scope of our results.

---

[4]See Appendix C for additional explanation about why this is the most general possible problem formulation.
[5]This corresponds to what is called an ergodicity assumption for stationary policies in Sutton and Barto [21]. See Appendix A for a more detailed discussion of this assumption and its implications.

**Dynamic Bayesian Network (DBN) Structure:** A Bayesian Network (BN) is a probabilistic graphical model that represents a set of variables and their conditional dependencies via a directed acyclic graph (DAG) and a dynamic Bayesian network (DBN) is a BN that relates variables to each other over adjacent time steps. In general, any Markov chain induced by a policy can be seen as a DBN over the state variables $s^1, ..., s^n$ that is also influenced by an action variable $a$ dictated by the policy [15]. The dependency graph $\mathcal{D}^\pi$ for the DBN is a directed cyclic graph whose nodes are $s^1, ..., s^n$ and where there is then a directed path from $s^i$ to $s^j$ in $\mathcal{D}^\pi$ iff $s_t^i$ influences $s_{t'}^j$ for some $t' > t$ under policy $\pi$. While we can ignore actions as nodes in the graph, it is still important to consider their effect in forming causal connections among the state variables. Note that every node in $\mathcal{D}^\pi$ should influence itself and thus have a self-loop. If there is a direct path from $s^i$ to $s^j$ and from $s^j$ to $s^i$, the variables $i$ and $j$ are said to be in the same strongly connected component. In contrast, if there is either a directed path from $s^i$ to $s^j$ or $s^j$ to $s^i$, but not both, the variables $i$ and $j$ are weakly connected such that the one that has a causal influence on the other can be considered as ordered before the one it causes. Let $\Gamma_1^\pi, ..., \Gamma_\ell^\pi$ be the maximal strongly connected components in $\mathcal{D}^\pi$ when an agent behaves with an arbitrary policy $\pi$ that is sorted such that if $i < j$ there are no directed edges from $\Gamma_j^\pi$ to $\Gamma_i^\pi$. We can then define $g^\pi = \max_i |\Gamma_i^\pi|$ as the maximum number of variables in any strongly connected component of $\mathcal{D}^\pi$ and $g = \max_\pi g^\pi$ as the maximum possible within a given policy class or parameterization $\pi$. In Appendix A.1 we provide examples to help readers better understand how this works and demonstrate that this formulation does not limit the scope of our results.

**Mixing Times and Coupling Times:** We can now leverage the well-studied connection between the mixing time and the so called *coupling time* to provide an upper bound on the mixing time of the Markov chain induced by $\pi$ [25]. Let $\tau$ be the random variable that represents the *coupling time* defined as the smallest $m$ for which two Markov chains starting from different initial states are in the same state. Following the analysis provided in Lemma 5.2 of [15], for any $\epsilon$, let $m$ be the number of steps such that for any two starting states in $\mathcal{S}$ if we can say that $P(\tau > m) \leq \epsilon$ then the Markov chain is $\epsilon$-mixed at time $m$. For this analysis we also must introduce the parameter $\beta_{i,\pi}$, which defines the minimum amount of common probability mass between any two state configurations for any state variable $i$ so that $\beta = \min_\pi \min_i \beta_{i,\pi} \in [0, 1]$ further corresponds to the minimum over all state variables $i$ and possible parameterizations of the policy $\pi$:

$$\beta_{i,\pi} = \min_{s_1, s_2 \in \mathcal{S}} \left( \sum_{s^i \in \mathcal{S}^i} \min \left\{ T^\pi(s^i|s_1), T^\pi(s^i|s_2) \right\} \right).$$

We can now introduce a mixing time bound as a function of the strongly connected component structure. While our result has similar motivation to Theorem 5.4 of Kearns and Koller [15], we note that the proof of Theorem 1 outlined in Appendix A is entirely our novel contribution building off Lemma 5.2 and Definition 5.3 of Kearns and Koller [15]. Additionally, there is no version of Kearns and Koller [15] online that includes the proof of Theorem 5.4.

---

**Theorem 1** *(Strongly Connected State Variables Bound): If the Markov chain $T^\pi(s'|s)$ induced by policy $\pi$ has $\ell$ maximal strongly connected components in $\mathcal{D}^\pi$ with a maximum size of $g$ state variables and a minimum of $\beta$ common probability mass between any two state configurations for any state variable, the mixing times $t_{ret}^\pi(\epsilon)$ and $t_{mix}^\pi(\epsilon)$ can be upper bounded:*
$$t_{ret}^\pi(\epsilon) \in t_{mix}^\pi(\epsilon) \in \mathcal{O}\left( \frac{1}{\beta^g} log(1/\epsilon) \right).$$

---

**Proof Sketch:** Due to the sorting of the strongly connected components, our analysis is based on coupling each of the $\Gamma_i^\pi$'s in succession. Because it is possible that multiple $\Gamma_i^\pi$'s couple at the same step, every step where the Markov chain does not fully couple must be a step where some $\Gamma_i^\pi$ does not couple. Our proof proceeds in the following high-level steps: 1) The probability of $\Gamma_i^\pi$ coupling at a given step once $\Gamma_1^\pi, ..., \Gamma_{i-1}^\pi$ have all already coupled is $\geq \beta^g$. 2) Thus the probability of $\Gamma_i^\pi$ not coupling at a step when $\Gamma_1^\pi, ..., \Gamma_{i-1}^\pi$ have all already coupled is $\leq (1 - \beta^g)$. 3) So the joint probability of $\Gamma_i^\pi$ not coupling for $m_i \geq 0$ steps when $\Gamma_1^\pi, ..., \Gamma_{i-1}^\pi$ have all already coupled is $\leq (1 - \beta^g)^{m_i}$. 4) If $\tau > m$ then $\sum_{i=1}^\ell m_i = m$ and the joint probability that $m$-steps have been spent not coupling in some $\Gamma_i^\pi$ has a probability bound independent of the particular allocation of $m$ into individual $m_i$. Thus we can conclude that $P(\tau > m) \leq (1 - \beta^g)^m$. 5) Leveraging the identity that $1 - x \leq e^{-x}$ for $x \geq 0$, we find that $P(\tau > m) \leq (e^{-\beta^g})^m$. 6) The Markov chain is $\epsilon$-mixed

if $P(\tau > m) \leq \epsilon$, so it must be $\epsilon$-mixed if $(e^{-\beta^g})^m \leq \epsilon$, which implies that $m \geq \frac{1}{\beta^g}log(1/\epsilon)$. 7) Finally, we note the relationship between $t^\pi_{ret}(\epsilon)$ and $t^\pi_{mix}(\epsilon)$ following Lemma 1 of [24].

**Bound Tightness:** Theorem 1 is a tighter version of Theorem 5.4 presented by Kearns and Koller [15] by a factor of $8\ell$ from their result of $\mathcal{O}\left(\frac{8\ell}{\beta^g}log(1/\epsilon)\right)$. This tighter bound underscores definitively that having more strongly connected components of a smaller size leads to tighter bounds on the mixing time than having fewer strongly connected components of a bigger size. In fact, without this contribution beyond the result by Kearns and Koller [15] it would not be possible to show our main result in Theorem 2 as it would thus be unclear if the mixing time bound gets tighter with the shrinking context length. An intuition we could provide for how this bound is achieved is that if the Markov chain has not coupled for $m$ steps, each time step must be spent not coupling in at least one of the strongly connected components, which has a bounded conditional probability for any component. Our detailed proof is in Appendix A. Moreover, while it may seem that only a restricted class of Markov chains have $\beta > 0$, in Corollary 2 of Appendix A, we demonstrate the extension of the definition for $\beta$ to the probability mass in common over $c > 1$ steps, which must be $> 0$ for some value of $c$ due to Assumption 1. Our approach could also yield an even tighter yet more cumbersome bound in terms of the maximum probability that any strongly connected component does not couple rather than using $\beta$ and $g$, which is important when we see significant variation in $\beta_i$ across variables.

**Conditioning on Variable Subsets:** One of the primary implications of Theorem 1 is that the mixing time of a policy is closely connected to the number of state variables that influence it's actions. For example, a policy in the class $\pi(\cdot|s)$, which takes full states $s = s^1, ..., s^n \in \mathcal{S}$ as input can be associated with $\beta_n$ and $g_n$. Meanwhile, a policy in an alternative class that only considers some potentially time dependent arbitrary subset of $n'(t) \leq n$ state variables as input at any moment in time $t$ such that $s^1, ..., s^{n'(t)} \subseteq s^1, ..., s^n$ can be associated with $\beta_{n'}$ and $g_{n'}$. Lemma 1 in Appendix A then shows that the mixing time bound gets strictly tighter because $1/\beta_{n'}^{g_{n'}} \leq 1/\beta_n^{g_n}$. We illustrate this phenomenon more concretely through two toy MDP examples detailed in Figure 1.

## 2.3 Building an Intuition with Examples

We will consider the mixing times for the two examples highlighted in Figure 1 over the space of deterministic policies. As in Riemer et al. [14], we relax the definition of the $\epsilon$-return mixing time using an average over start states (rather than a maximum) in order to emphasize mixing times encountered in practice. See Appendix B for details on our methodology for estimating mixing times.

**MDPs with Irrelevant Variables (Figure 1):** In the first example we consider, there are two independent state variables $x$ and $y$ that are both influenced by the agent's actions. As a result, $x$ and $y$ are in the same strongly connected component if both variables also influence the agent's actions. Because the reward only depends on $x$, the agent can achieve the optimal policy without considering the $y$ variable in its decision about actions. If the policy is only conditioned on the variable $x$, $x$ influences $y$ through its actions, but $y$ no longer has an influence on $x$. As a result, the Markov chain induced by the policy that only conditions on the relevant variable $x$ has two strongly connected components of size one such that $g = 1$ whereas the policy that conditions on both relevant and irrelevant variables potentially has a single strongly connected component of size two such that $g = 2$ with $\beta$ held constant. Based on Theorem 1, we expect the policy conditioned on both variables to experience higher mixing times even despite the small scale of this toy problem. Indeed, this is the case when we compute the $\epsilon$-return mixing time at a reward rate precision of $\epsilon = 0.05$, which must reflect a minimum of 10% relative estimation error because no policy has a reward rate of 0.5. The optimal policy has a mixing time of 17.6 and only depends on the variable $x$. In fact, no policy only depending on $x$ has a mixing time above 44.9. Meanwhile, there is a policy dependent on both variables that is within the reward rate precision threshold $\epsilon$ of the optimal policy with a mixing time of 45.2, and a policy conditioned on both variables with a mixing time as high as 340.3.

**Impact of Reward Density:** To test if a tighter result is possible with respect to the ratio between the maximum mixing times predicted by Theorem 1, we have tried a setting where the reward is still 1.0 when $x = x0$, but edited to $-0.25$ when $x = x1$ and $-1.5$ when $x = x2$ rather than 0. This setting was chosen to have no impact on the optimal policy while densifying rewards without making a substantial change to the magnitude of best reward rates achieved. The ratio between the maximum mixing times is now improved to 8.8 times smaller (233.0 vs. 2049.7) when only focusing on the

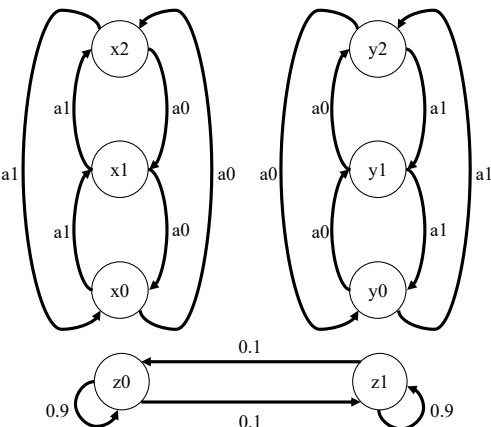

Figure 1: **Illustrative Toy Examples.** The above figure details the three relevant variables $x$, $y$, and $z$ that we will consider for our toy examples. Note the action $a0$ is interpreted as $a1$ or vice versa with a 10% failure probability, which is the same as the rate at which $z$ switches regardless of the agent's actions. **Irrelevant Variables Example:** In this case, we consider only the variables $x$ and $y$ ($z$ is not needed in this example) where the reward is $+1$ if $x = x0$ and 0 otherwise. The result is that variable $x$ is relevant to the task whereas variable $y$ is irrelevant. **Independent Subtasks Example:** In this case, we consider that the variable $x$ evolves (according to the diagram) with $y$ remaining constant when $z = z0$ and the variables $y$ evolves with $x$ remaining constant when $z = z1$. The reward is $+1$ if $z = z0$ and $x = x0$ or if $z = z1$ and $y = y0$, or 0 otherwise.

relevant variables rather 7.6 times smaller (44.9 vs. 340.3) with the previous sparse rewards. Note that the reward still has no dependence on $y$ and is thus invariant to a large part of the change in the state. So, mixing is still substantially faster with respect to the reward ($t_{\text{ret}}^{\pi}(\epsilon)$) than the state ($t_{\text{mix}}^{\pi}(\epsilon)$).

**MDPs with Independent Subtasks (Figure 1)**: We can now consider an example with three variables $x$, $y$, and $z$ where the agent's actions influence both $x$ and $y$. A policy conditioned on all three variables has a strongly connected component of size one including $z$ (because $z$ is not influenced by the agents actions or $x$ or $y$) and a strongly connected component of size two including $x$ and $y$ such that $g = 2$. On the other hand, if the policy were able to condition only on $x$ when $z = z0$ and only condition on $y$ when $z = z1$, then the actions that still influence both $x$ and $y$ no longer serve to make their values dependent on each other. This results in three strongly connected components of size one such that $g = 1$ with $\beta$ held constant. One for each variable $x$, $y$, and $z$. Based on the analysis of Theorem 1, we would again expect the policy conditioned on all variables always to experience higher mixing times than the policy conditioned on only the relevant variables for the specific subtask even despite the small scale of this toy problem. As for the previous example, we compute the $\epsilon$-return mixing time at a reward rate precision of $\epsilon = 0.05$, reflecting a minimum of 10% relative estimation error. The optimal policy has a mixing time of 16.0 and only depends on the variable $x$ when $z = z0$ and variable $y$ when $z = z1$. In fact, no policy structured like this has a mixing time above 267.1. Meanwhile, there is a policy dependent on all variables that is within the reward rate precision threshold $\epsilon$ of the optimal policy with a mixing time of 341.2. Moreover, there is a policy conditioned on all variables with a mixing time as high as 968.1. This example helps illustrate the important fact that a policy can limit its mixing time while still conditioning on all variables if the subset it conditions on has some time dependence.

## 3 Understanding How the Context Length Impacts the Mixing Time

In the previous section we were concerned only with fully observable MDPs, but real-world and large scale problems tend to be partially observable. We will begin by highlighting new notation that allows us to talk about notions of optimality as a function of the interaction history context length and present mixing time bounds that are dependent on the context length considered by a policy. We go on to conduct experiments that verify the relevance of this analysis during online RL in partially observable domains and demonstrate that Transformer neural network models actually experience higher mixing times than alternative function approximators and tabular models.

### 3.1 Partially Observable Environments with Local Observation Structure

**Partially Observable Environments:** We can extend the notion of an MDP from the previous section to consider an unknowable but ultimately stationary Partially Observable Markov Decision Process (POMDP) [26], which is comprised of an augmented tuple $\mathcal{P} \doteq \langle \mathcal{S}, \mathcal{A}, \mathcal{O}, R, T, O \rangle$. This adds to the definition of an MDP $\mathcal{M}$ an observation space $\mathcal{O}$ and an observation function $O(o|s)$ that maps states $s \in \mathcal{S}$ to observations $o \in \mathcal{O}$.[6] The state $s \in \mathcal{S}$ of such a POMDP ultimately serves as a theoretical quantity that is never actually observed from the agent's perspective and the observations $o \in \mathcal{O}$ that it does receive may have an arbitrarily non-Markovian relationship.

**Interaction Histories:** At time $t$ the union of all things the agent has observed about an environment can be called its interaction history $h_t := \{o_0, a_0, r_0, o_1, a_1, r_1, ..., o_t\}$.[7] We can also consider the case where the agent only maintains a finite window of history with size $k$ i.e., $h_t^{(k)} := \{o_{t-k+1}, a_{t-k+1}, r_{t-k+1}, x_{t-k+2}, a_{t-k+2}, r_{t-k+2}, ..., o_t\}$ where our notation is chosen so that a memoryless policy that only processes the current observation corresponds to $k = 1$. We will henceforth call $k$ the context length of the interaction history. A given POMDP $\mathcal{P}$ is non-Markovian to order $k_\mathcal{P}$ which implies that for every combination of state $s$, action $a$, and history window $h^{(k)}$, $T(\cdot|s, a) = T(\cdot|h^{(k)}, a)$ and $R(\cdot|s, a) = R(\cdot|h^{(k)}, a) \ \forall k \geq k_\mathcal{P}$.

**Optimality in POMDPs:** An advantage of using a policy that processes the full interaction history is that it is possible to handle arbitrarily non-Markovian environments, but this is not computationally scalable in the long-run. As $t \to \infty$, $\pi_\theta(\cdot|h_t)$ processes an infinite length sequence $h_t$. In practice, we must limit the context length sent to the policy. The best policy $\pi_\theta(\cdot|h_t^{(k)})$ over a finite context length has the same optimal reward rate as the best policy $\pi_{\theta'}(\cdot|s_t)$ over true state inputs when $k \geq k_\mathcal{P}$ such that $\lim_{t\to\infty} \max_{\theta \in \Theta} \mathbb{E}_{\pi_\theta(\cdot|h_t^{(k)})}[r_t] = \lim_{t\to\infty} \max_{\theta' \in \Theta'} \mathbb{E}_{\pi_{\theta'}(\cdot|s_t)}[r_t]$.[8] However, this implies the context length must scale with $k_\mathcal{P}$, which could become quite large in highly non-Markovian real-world environments where achieving optimality is difficult.

**The Effect of Context Length:** As described in the previous section, we are interested in multi-dimensional state spaces consisting of $n$ variables and in particular problems where $n$ is large. A given observation $o$ generated from $O(o|s)$ may be then caused by only a subset of the state variables $Par(o) \subseteq \{1, .., n\}$ where $Par(o)$ denotes the causal parent state variables of $o$. As such, a finite history window $h^{(k)}$ may also be caused by a subset of the state variables because the same can be said for rewards and actions that are potentially included in this input. In general, a finite context length is reflective of a maximum $n_k(t)$ sized subset of the state variables at a time $t$ where $n \geq n_{k'}(t) \geq n_k(t)$ if context length $k' \geq k$ for all $t$. This is because $Par(h^{(k)}) \subseteq Par(h^{(k')}) \subseteq \{1, .., n\}$. Theorem 2 then follows from the combined results of Lemma 1 in Appendix A and Theorem 1.

---

**Theorem 2** *(Limiting Mixing Times with Context Length): If the Markov chain induced by a policy conditioned on a finite interaction history window $\pi(\cdot|h^{(k)})$ with a context length of $k$ has $\ell_k$ maximal strongly connected components in $\mathcal{D}^\pi$ with a maximum size of $g_k$ variables and a minimum of $\beta_k$ common probability mass between any two state configurations, the mixing times $t_{ret}^\pi(\epsilon)$ and $t_{mix}^\pi(\epsilon)$ can be bounded for any $k' \geq k$:*

$$t_{ret}^\pi(\epsilon) \in t_{mix}^\pi(\epsilon) \in \mathcal{O}\left(\frac{1}{\beta_k^{g_k}} log(1/\epsilon)\right)$$

$$\in \mathcal{O}\left(\frac{1}{\beta_{k'}^{g_{k'}}} log(1/\epsilon)\right)$$

*where the dependence on the context length implies that $0 \leq \beta_{k'} \leq \beta_k \leq 1$ and $g_{k'} \geq g_k \geq 1$.*

---

**Proof Sketch:** Lemma 1 in the appendix considers the mixing time relationship of policy classes conditioned on subsets of the state variables that other policy classes are conditioned on. Our

---

[6] Note that any non-stationary MDP can alternatively be viewed as a POMDP (see Proposition 2 of Khetarpal et al. [27]) and by the same logic any non-stationary POMDP can alternatively be viewed as a stationary POMDP, so there is no loss of generality in making the assumption that the POMDP is stationary.

[7] Many exclude reward from the history representation in POMDPs, but we include it to be as general as possible. That said, our theoretical results do not require this particular choice of the history representation.

[8] Note the distinction between $\Theta'$, the possible policies over $s_t$, and $\Theta$, the possible policies over $h_t^{(k)}$.

proof includes the following high-level steps by applying our notation in which $k' \geq k$ for all $t$ to the results of Theorem 1 and Lemma 1: 1) We consider the causal parent state variables of each observation, action, and reward to conclude that $Par(h^{(k)}) \subseteq Par(h^{(k')}) \subseteq \{1, .., n\}$, which implies that $n \geq n_{k'}(t) \geq n_k(t)$. 2) Through Lemma 1 we show that by rule of Cartesian products over subsets that $0 \leq \beta_{k'} \leq \beta_k \leq 1$. 3) Through Lemma 1 we also demonstrate that $g_{k'} \geq g_k \geq 1$ because causal connections in $\mathcal{D}^\pi$ are only added and not removed when the context length is increased. 4) This then implies that $1/\beta_{k'}^{g_{k'}} \geq 1/\beta_k^{g_k}$, which is sufficient to prove Theorem 2 using Theorem 1 because $\epsilon$ is independent of $k$.

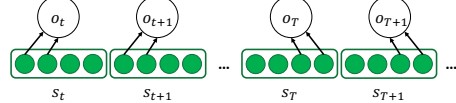

**Formalizing the Tradeoff:** Theorem 2 formally establishes a novel connection between the context length $k$ of a policy and bounds on its mixing time. Growing $k$ increases the performance of the best achievable policy when $k_{\mathcal{P}}$ is high. However, any increase in $k$ leads to a looser mixing time bound.

**When it Really Matters:** While this bound is always true, there may not be a meaningful dependence on $k$ for some problems. This includes very simple environments where $k_{\mathcal{P}} \to 1$ or environments with no local structure where all state variables influence all observations. However, if changing the context length $k$ does actually lead to a change in the strongly connected component structure, the impact on the mixing time will be exponential. In Figure 2 we present an illustration of the kinds of problems where the chosen value of $k$ may make a large difference in the value of $n_k(t)$. We can see that at each time only a subset of the total state variables contributes to the local observations of the agent and that which variables they are has a degree of local consistency across time-steps of the Markov chain induced by the policy in the environment. When observations are only caused by a subset of the state variables at each step, there is more potential to break the problem up into independent subtasks as in our example in the previous section.

Figure 2: **POMDPs with Local Observation Structure.** An example of an environment with a multidimensional state space where near term observations are only causally dependent on a subset of the variables. At time $T >> t$ we see different dimensions of the state space influencing observations and thus these separate subsets do not need to be in the same strongly connected component.

### 3.2 Empirical Verification During Online RL

**Context Length and Encountered Mixing Times:** While Theorem 2 draws a clear connection between context length and mixing times, there still remains a question about if this analysis is too conservative and it is unclear if these policies will actually be encountered during learning. To test this question empirically, we consider the simple RGB world environment in Figure 6a, which is modeled after the classic T-maze [28] environment. We conducted comprehensive experiments leveraging tabular Q-learning for 1 billion steps over 100 seeds (see Appendix B). In Figure 3 we highlight that longer context lengths experience larger average mixing times at the same approximate reward rate as policies conditioned on lower context lengths during learning. When interpreting this kind of plot, it is important to consider that there is a difference between the input required to express a policy and the input available

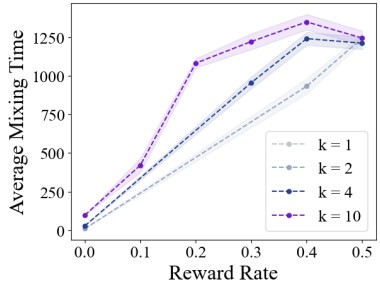

Figure 3: **Mixing Times Encountered vs. Context Length.** We plot the average mixing time and 95% confidence intervals encountered during 1 billion steps of learning over 100 seeds at each context length $k$. We bin the average mixing time computation by the nearest 0.1 increment of the reward rate of the policy. $k = 1$ is not visible because the reward rate is always 0.

to that policy. Any policy that receives a sufficient statistic of the environment state as input i.e. $k \geq 2$ will arrive at the same optimal policy by the end of training, so what is more interesting is the points encountered along the way.

**Effect of Neural Network Architecture:** We extend these experiments to prominent neural network architectures on the same problem leveraging Q-learning (see Appendix B). We plot the result when each architecture has about 1M parameters in Figure 4. All models have the same mixing time when they arrive at the same policy i.e. the optimal policy. However, at intermediate reward rates we see a higher average mixing time for Transformer models. It is important to note that sometimes an architecture will learn to achieve a reward rate that others never learn to achieve. In this case, there

is not a clear basis for comparison. We also ran experiments evaluating the role of the number of parameters. The extra capacity did not have a statistically significant effect on the mixing times for MLP and LSTM models. Meanwhile, extra capacity resulted in higher encountered mixing times throughout learning and a larger effect when increasing $k$ for Transformer models (see Figure 8). It appears that attention mechanisms make it easier to focus on the full context rather than i.e. only the recent parts and that this capability is predictably enhanced when the model capacity is increased. In Appendix B we take a closer look at the Transformer attention maps in the decoder and present some evidence that the larger models are paying attention more uniformly to the entire context.

## 4 Understanding Growing Context Lengths in Foundation Models for RL

**Foundation Models for RL:** Large scale foundation models trained to recreate behaviors from a large and diverse distribution have recently had a disrupting effect across the field of AI. Concretely, a foundation model leverages an offline dataset $\mathcal{D}$, containing data from $N$ different now unknown behavior policies $\pi_i$ for $i \in \{1, ..., N\}$. Decision Transformers [19] is a popular approach that achieved state of the art performance in offline RL by treating learning as a sequence modeling problem akin to language modeling. The objective is to imitate the behavior of all policies $\pi_i$ in the dataset using a window of their interaction history $h^{(k)}$. See Appendix B for additional details.

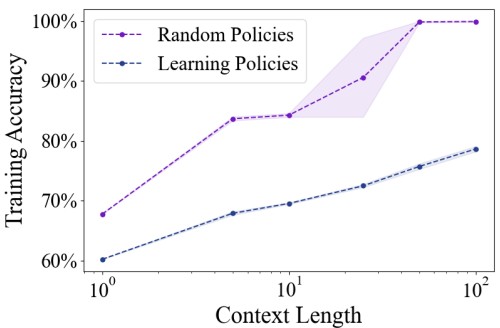

Figure 4: **Policy Architecture vs. Encountered Mixing Times.** We plot the average mixing time and 95% confidence intervals for each choice of policy architecture between tabular, MLP, LSTM, and Transformer models with averages binned by the reward rate. We provide a representative example with $k = 5$ and include comprehensive plots in Appendix B. Mixing times goes down as the absolute value of the reward rate gets close to 0 as a consequence of the sparse and local reward structure.

**Fitting Training Data:** Foundation models may struggle to model the training data when the context window length $k$ used for training is less than the maximum context window of any policy $k_{\mathcal{D}} := \max_{i \in \{1,...,N\}} k_i$. The reason is because our model must not only produce the behavior of each agent $\pi_i$, but also, must store additional context to disambiguate $\pi_i$, from all $\pi_j \; \forall j \in \{1, ..., N\} \setminus i$. Thus it will be necessary to consider large contexts, well beyond that of any context length used by the behavior policies, especially as the number and diversity of behavior policies considered grows.

**Comparison to Behavior Policy Context Lengths:** To understand the connection between the behavior policy context length and the context length needed to train Decision Transformers, we consider a similar setup to the Key-Door experiments in the original paper [19], but with the publicly available Minigrid Crossing environment [29] (Figure 6b). We randomly initialized 1,000 deterministic behavior policies, each rolled out for one episode in the environment to collect dataset $\mathcal{D}$ leveraging a CNN architecture following past work on the Minigrid domain [30; 31]. The context length of each behavior policy is set to 1, implying $k_{\mathcal{D}} = 1$. In Figure 5 we plot the training accuracy achieved by a Decision Transformer model as a function of its context length $k$. We report that the training accuracy is only optimal in general for $k \geq 50$, which is significantly higher than $k_{\mathcal{D}} = 1$ with $k = 25$ reaching optimal performance for some random seeds. We also considered 1,000 episodes generated by different stochastic behavior

Figure 5: **Context Length vs. Training Accuracy.** We plot the achieved training accuracy and 95% confidence intervals across 5 random seeds as a function of the Decision Transformer context length in the crossing environment with 1,000 episodes of data generated by either random behavior policies with a context length $k = 1$ or a REINFORCE based learning agent using a context length $k = 1$.

policies of REINFORCE based learning agents [32] that learn following every episode. Figure 5 demonstrates that it is even harder to model these policies. This makes sense both because the policy distinctions are more subtle and less diverse between episodes and because the policies are stochastic, leaving an irreducible source of uncertainty.

**Evaluating Learned Models:** It could be argued that, although a larger context length may be required to capture the complete range of behavior policies compared to each policy individually, this merely promotes overfitting of the training data and does not effectively enhance downstream performance. So, to see if Decision Transformers need near optimal training accuracy for good downstream performance, we evaluated each Decision Transformer model for a particular random policy seed where $k = 25$ is able to achieve optimal training performance across a variety of return to go prompts stepping by $0.01$ from $0$ to $1.0$ and report the average performance across 1,000 episodes. We plot our results in Figure 9 which validate the importance of fitting the training data. Decision Transformer with $k = 25$ can achieve performance as good as any behavior policy used to generate the data. Meanwhile, models with smaller $k$ fail to achieve the same performance. Moreover, increasing context length $k$, while important for optimization, raises the mixing time. The average mixing time with precision $\epsilon = 0.01$ across policies and start states for the Decision Transformer with $k = 25$ is 298.1 episodes or 86,834.3 steps while the average mixing time is merely 11.2 episodes or 3,271.2 steps for the behavior policy across policies and start states. Figure 9 also demonstrates the not very surprising conclusion that unnecessarily large context lengths are more prone to overfitting than the minimal context length that achieves 100% training accuracy.

# 5    Discussion and Future Work

In this work, we have highlighted the potential limitations of training models conditioned on ever increasing context lengths and particularly the effect that these growing context lengths have on mixing times. This motivates a number of interesting research questions to explore moving forward.

**New Architectures and Algorithms:** Most work on RL that even acknowledges the challenges associated with high mixing times does so with a defeatist mentality, assuming that problems with high mixing times are unavoidably harder and that there is basically nothing that could be done about it. Our work highlights that this isn't actually true and that the policy class we choose to optimize over itself can have a big impact on mixing properties. What our paper shows in Theorem 2 is that what leads to potentially high mixing times is when our model leverages a monolithic representation that is highly sensitive to a large part of the interaction history at all times. This is particularly descriptive of how vanilla transformers work, but there are multiple already existing research directions that seem well suited to scaling to high context lengths while providing less history sensitivity at each step. We refer interested readers to Appendix D for an in depth discussion of related directions.

**Scaling to Complex Environments:** We believe the settings of greatest relevance to our work are those related to continual or multi-task environments where agents are evaluated as generalists over a number of skills rather than just solving a single narrow task. As such, we believe that focus on the difficulties presented by high mixing times is timely in the age of foundation models. As mentioned at the end of Section 3.1, our analysis will not have relevance in problems where there are few state variables that each impact every observation. However, composite tasks that test a number of sub-skills naturally tend to have many total state variables with relatively few impacting each observation. So, for example, simple Atari domains will not suffer from high mixing times, but i.e. continual learning over multiple Atari games will [14] as a result of the sparsity of causal impact of variables across games. Broadly speaking, AI assistant tasks that include providing help on a number of topics rather than just one should also suffer from issues with high mixing times.

**Evaluation of Foundation Models:** Our work additionally highlights how the way we pretrain foundation models may make high confidence evaluation of these models more difficult. Trusted evaluation of foundation models remains an important challenge for the research community to grapple with. Towards this end, our work is the first to establish that more interaction is needed to reliably evaluate models that have larger context lengths. This novel perspective is very important for researchers to consider as these models are increasingly being deployed in the real-world.

## Acknowledgments and Disclosure of Funding

We would like to thank Murray Campbell, Miao Liu, and Payel Das for valuable conversations over the course of this project. This project was supported by the IBM-Mila collaboration grant. We would also like to acknowledge our support from the Canada CIFAR AI Chair Program and from the Canada Excellence Research Chairs (CERC) Program. We thank the IBM Cognitive Compute Cluster and the Mila cluster for providing computational resources. Finally, we really appreciate the feedback we received from the NeurIPS reviewers, which helped to improve the presentation of our key findings.

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

# A    Deriving Upper Bounds for the Mixing Time

**Section Overview:** In this section, we provide proofs for Theorem 1 and Theorem 2 in the main text. We also highlight Corollary 2, which is a key extension of Theorem 1 that relaxes the definition of $\beta$ to be over $c$-steps rather than a single step. Moreover, we highlight Lemma 1, which is a key intermediate result to bridge the gap between the results in Theorem 1 and Theorem 2. For each theoretical result, we first state the result and then provide a detailed proof. Our proofs build off the definitions and notation established in the main text.

For clarity, we will briefly recap the **key assumptions** discussed in the main text and why each is needed:

1. **Unichain Policies:** All stationary policies are aperiodic and unichain, meaning they give rise to a Markov chain with a single recurrent class that is recurrent in the Markov chain of every policy.

   - Needed for Theorem 1, Corollary 2, Lemma 1, and Theorem 2.
   - This assumption is necessary given our definitions of the mixing time, the average reward, and coupling time in the main text. For example, it is otherwise possible that two copies of the Markov chain originating from different start states may never couple (even in the infinite limit).

2. **Multidimensional State Space:** Each state is separated into $n$ variables i.e., $s = [s^1, ..., s^n]$ with $s^i \in \mathcal{S}^i$ for all variables $i \in \{1, ..., n\}$ such that $\mathcal{S} \subseteq \mathcal{S}^1 \times ... \times \mathcal{S}^n$ and $T(s'|s, a) = \Pi_{i \in \{1,..,n\}} T(s'^i|s, a)$.

   - Needed for Theorem 1, Corollary 2, Lemma 1, and Theorem 2.
   - This assumption is necessary to make it meaningful to talk about the Markov chain induced by a policy as a DBN over state variables and the strongly connected component structure of such a DBN.

3. **History Windows Depend on Subsets of State Variables:** Observation $o$, action $a$, and reward $r$ are caused by of a subset of the state variables such that $\text{Par}(o) \subseteq \{1, .., n\}$, $\text{Par}(a) \subseteq \{1, .., n\}$, and $\text{Par}(r) \subseteq \{1, .., n\}$. As a result, any interaction history window $h^{(k)}$ of size $k$ is also caused by a subset of the state variables such that $\text{Par}(h^{(k)}) \subseteq \{1, .., n\}$.

   - Needed for Theorem 2 only because it is the only result in this section concerned with policies over $h^{(k)}$.
   - This assumption is not strictly necessary to state in the sense that it also covers the corner case where $h^{(k)}$ depends on all $n$ variables and thus must always hold in practice. However, it is worth emphasizing because if we do not actually see a change in the causal subset of variables as $k$ decreases, then our bounds on the mixing time do not actually get tighter but just stay the same. As a result, this helps us build an intuition for the kinds of problems where the bound tightening showcased in Theorem 2 is meaningful (even though the result of Theorem 2 is still valid as stated even when it is not).

## A.1    Understanding the Generality of our Assumptions

**Multidimensional State Space:** The assumption that there are $n$ state variables $s = [s^1, ..., s^n]$ subsumes the corner case where $n = 1$ leaving only one variable. Moreover, the assumption that $T(s'|s, a) = \Pi_{i \in \{1,..,n\}} T(s'^i|s, a)$ also is not restrictive because it is only required that this structure hold for some representation of the state variables. For example, if two variables $s^a \in \mathcal{S}^a$ and $s^b \in S^b$ do not involve independently of each other, a new variable can be made to represent the

combination of these variables $s^c \in S^c := S^a \times S^b$. In the worst case, the system reduces to $n = 1$ where there is a single discrete variable evolving on its own. If this is the case, it does not invalidate the theoretical results of Theorem 1 and Theorem 2. However, as explained in the paragraph on *When it Really Matters* in Section 3, when $n = 1$ the bound on the mixing time does not change with $k$ as there is only one strongly connected component and the inequality from both theorems is simply an equality. As such, our results are most meaningful in environments with local observation structure, where are actually quite common for real-world and large-scale applications.

**DBN Structure:** Multidimensional transition functions can be represented by set of DBNs including a separate DBN for each action [33]. The DBN for any action $a \in \mathcal{A}$ is an acyclic directed graph with two layers. The first layer is the set of state variables in the current state and the second layer is the set of state variables in the next state [34]. The Markov chain induced by a given policy can be seen as DBN formed as a mixture of the DBNs for each action according to the policy and with the actions produced by the policy also critically being causally dependent on certain state variables. The assumption that there are $\ell$ strongly connected components $\Gamma_1^\pi, ..., \Gamma_\ell^\pi$ is also not restrictive because $\ell = 1$ is a corner case of this formulation. Indeed, the generality of this phenomena underlies the generality of planning algorithms such as Topological Value Iteration (TVI) [35], which efficiently applies Value Iteration (VI) on each strongly connected component in reversed topological order. TVI is indeed known to cover the case when $\ell = 1$ [34]. Again, the case where $\ell = 1$, which also happens by definition whenever $n = 1$, is not very interesting for our theory, but does not invalidate our results.

**Motivating Examples:** To motivate the generality of these assumptions in real-world settings, we will walk through how both of the domains used in our experiments naturally possess the desired structure to make our bounds in Theorem 1 and Theorem 2 meaningful despite their simplicity. See Figure 6 for visual depictions of the two domains.

- **RGB World (Figure 6a):** This domain can be represented as a multidimensional state space with the RGB color of each cell in the line representing its own state variable. The colors of each state variable is only possibly influenced by the variable on either side of it (depending on the chosen action of the agent). As a result, it is clear that the variables in the transition graph can be considered to evolve independently of each other given the previous state and action while displaying sparse interaction structure among variables. As such, the number of strongly connected components is highly dependent on the policy parameterization. Moreover, the observations are local to the agent, and are only the causal result of the agent's surrounding state variables in a single observation with multiple observations needing to be combined to reflect the full state space.

- **Minigrid Crossing (Figure 6b):** This domain can be represented as a multidimensional state space with each grid cell representing its own state variable. Each variable can either be the location of the agent, a wall, an empty space or the location of the goal. Each variable can then only be influenced by the variables adjacent to it across a single time step. Thus the the variables in the transition graph can be considered to evolve independently of each other given the previous state and action while displaying sparse interaction structure among variables. As a result, the number of strongly connected components is yet again highly dependent on the policy parameterization. The observations are also yet again local to the agent, and only are causally dependent on the state variables within a limited radius of the agent's location. To reflect all state variables, it is indeed necessary that multiple observations are combined.

### A.2 Proof of Theorem 1

**Theorem 1** (*Strongly Connected State Variables Bound*): *If the Markov chain $T^\pi(s'|s)$ induced by policy $\pi$ has $\ell$ maximal strongly connected components in $\mathcal{D}^\pi$ with a maximum size of $g$ state variables and a minimum of $\beta$ common probability mass between any two state configurations for any state variable, the mixing times $t_{ret}^\pi(\epsilon)$ and $t_{mix}^\pi(\epsilon)$ can be upper bounded:*

$$t_{\text{ret}}^\pi(\epsilon) \in t_{\text{mix}}^\pi(\epsilon) \in \mathcal{O}\left( \frac{1}{\beta^g} log(1/\epsilon) \right).$$

**Proof Sketch:** Due to the sorting of the $\ell$ strongly connected components, our analysis is based on coupling each of the $\Gamma_i^\pi$'s in succession. Because it is possible that multiple $\Gamma_i^\pi$ couple at the same step, every step where the Markov chain does not fully couple must be a step where some $\Gamma_i^\pi$ does not couple. Our proof proceeds in the following high-level steps:

1. The probability of $\Gamma_i^\pi$ coupling at a given step once $\Gamma_1^\pi, ..., \Gamma_{i-1}^\pi$ have all already coupled is $\geq \beta^g$.

2. Thus the probability of $\Gamma_i^\pi$ not coupling at a step when $\Gamma_1^\pi, ..., \Gamma_{i-1}^\pi$ have all already coupled is $\leq (1 - \beta^g)$.

3. So the joint probability of $\Gamma_i^\pi$ not coupling for $m_i \geq 0$ steps when $\Gamma_1^\pi, ..., \Gamma_{i-1}^\pi$ have all already coupled is $\leq (1 - \beta^g)^{m_i}$.

4. If $\tau > m$ then $\sum_{i=1}^\ell m_i = m$ and the probability that $m$-steps have been spent not coupling in some $\Gamma_i^\pi$ has a probability bound independent of the particular allocation of $m$ into $m_i$. Thus we can conclude that $P(\tau > m) \leq (1 - \beta^g)^m$.

5. Leveraging the identity that $1 - x \leq e^{-x}$ for $x \geq 0$, we find that $P(\tau > m) \leq (e^{-\beta^g})^m$.

6. The Markov chain is $\epsilon$-mixed if $P(\tau > m) \leq \epsilon$, so it must be $\epsilon$-mixed if $(e^{-\beta^g})^m \leq \epsilon$, which implies that $m \geq \frac{1}{\beta^g} log(1/\epsilon)$.

7. Finally, we note that $t_{\text{ret}}^\pi(\epsilon) \in t_{\text{mix}}^\pi(\epsilon)$ following Lemma 1 of Kearns and Singh [24].

**Additional Details:** Due to the sorting of the $\ell$ strongly connected components, our analysis will be based on stabilizing the $\Gamma_i^\pi$'s in succession as in Kearns and Koller [15]. For example, if we assume that $\Gamma_1^\pi, ..., \Gamma_{i-1}^\pi$ have all stabilized by time $t$, all the variables in $\Gamma_i^\pi$ must then couple at the same time for $i$ to stabilize. This event then happens at time $t$ with probability $\geq \beta^g$. As soon as $i$ stabilizes, we can move on to stabilizing $i + 1$. When all $\ell$ strongly connected components have stabilized, we are done and have surpassed the coupling time. An important subtlety to note is that this process need not take at least $\ell$ steps and will in fact complete in just a single step with probability $\geq (\beta^g)^\ell$. It is not that a strongly connected component $i$ can't couple until $i - 1$ has coupled at the previous step, but rather that if $i$ has coupled and $i - 1$ has not yet stabilized, this coupling over a subset of variables is not necessarily meaningful for $i$ stabilizing.

To formalize this idea, let us consider that each strongly connected component $i$ was the focus of the successive stabilization for $m_i \geq 0$ steps without stabilizing yet. If the entire Markov chain across all $\ell$ components has been running for $m$ steps and has not yet fully stabilized, we note that $\sum_{i=1}^\ell m_i = m$. This is because every step where a component $i$ did stabilize, either the next component $i + 1$ did not stabilize at the same step, or we consider a series of components $j > i$ where each stabilizes immediately i.e. $m_j = 0$ until the next component $j + 1$ does not stabilize. Note that if only $i - 1$ components have stabilized after $m$ steps that implies that $m_j = 0$ for all $j > i$, so this still describes the setting where not all components actually have the opportunity to stabilize despite $m_i$ being defined for all $i \in \{1, ..., \ell\}$. The probability that the system has not stabilized by $m$ steps $P(\tau > m)$ is then equal to the probability across all allocations of $m_i$ that each component $i$ did not couple for $m_i \geq 0$ steps when all components $j < i$ had already stabilized. The probability of $i$ stabilizing at a given step if $i - 1$ also stabilized at or before that step is $\geq \beta^g$, so the probability of not stabilizing at a given step is $\leq (1 - \beta^g)$, and the probability of not stabilizing for each of $m_i$ steps is $\leq (1 - \beta^g)^{m_i}$. Then we can consider the joint probability of this across all $\ell$ components to upper bound $P(\tau > m)$ noting that the total probability of all possible allocations of $m_i$ must be $\leq 1$:

$$P(\tau > m) \leq \prod_{i=1}^\ell (1 - \beta^g)^{m_i} = (1 - \beta^g)^{\sum_{i=1}^\ell m_i} = (1 - \beta^g)^m \leq (e^{-\beta^g})^m. \tag{1}$$

The dependence between the strongly connected components in the joint probability is incorporated through interdependence between the $m_i$ allocations in the product. However, we see that this interdependence and dependence on the number of strongly connected components goes away when noting that $\sum_{i=1}^\ell m_i = m$. In the final inequality of the above equation, we note that $1 - x \leq e^{-x}$

for $x \geq 0$. The system is $\epsilon$-mixed after $m$ steps if $P(\tau > m) \leq \epsilon$ therefore by the transitive property it is then also $\epsilon$-mixed if:

$$(e^{-\beta^g})^m \leq \epsilon. \tag{2}$$

We then proceed by taking the logarithm of both sides of this equation:

$$m(-\beta^g) \leq log(\epsilon). \tag{3}$$

We can then multiply each side by negative one, which also reverses the inequality:

$$m(\beta^g) \geq log(1/\epsilon). \tag{4}$$

Finally, we divide both sides by $\beta^g$ and yield the result that it must be $\epsilon$-mixed if:

$$m \geq \frac{1}{\beta^g} log(1/\epsilon). \tag{5}$$

This then implies that:

$$t_{\text{mix}}^\pi(\epsilon) \in \mathcal{O}\left( \frac{1}{\beta^g} log(1/\epsilon) \right) \tag{6}$$

The final step of the proof is simply to note that the result from Lemma 1 of Kearns and Singh [24] that $t_{\text{mix}}^\pi(\epsilon) \in \Omega(t_{\text{ret}}^\pi(\epsilon))$ for any policy $\pi$.

## A.3 Proof of Corollary 2

**Corollary 2** *(Strongly Connected State Variables Bound with $c$-Step Transitions): If the Markov chain $T^\pi(s'|s)$ induced by policy $\pi$ has $\ell$ maximal strongly connected components in $\mathcal{D}^\pi$ with a maximum size of $g$ state variables and a minimum of $\beta_c$ common probability mass between any two state configurations for any state variable after $c \geq 1$ steps, the mixing times $t_{ret}^\pi(\epsilon)$ and $t_{mix}^\pi(\epsilon)$ can be upper bounded:*

$$t_{\text{ret}}^\pi(\epsilon) \in t_{\text{mix}}^\pi(\epsilon) \in \mathcal{O}\left( \frac{c}{\beta_c^g} log(1/\epsilon) \right)$$

A potential criticism of $\beta$ from [15] is the potential for $\beta = 0$ when there are deterministic elements of the transition dynamics. However, this concept could be relaxed by considering the $c$-step transition dynamics instead of just 1 step. For policies following Assumption 1, there must be a common recurrent set of states that all policies experience regardless of the starting state, so values must become greater than 0 for appropriately large values of $c$. As such, we define $\beta_c$ a parameter that defines the minimum amount of common probability mass for any two state configurations for any state variable after $c$-steps so that $\beta_c = \beta_{i,c} \in [0, 1]$:

$$\beta_{i,c} = \min_{u,u' \in \mathcal{S}} \left( \sum_{s^i \in \mathcal{S}_i} \min(T^\pi(s^i|u, c), T^\pi(s^i|u', c)) \right) \tag{7}$$

It is easy to show this because the probability of a strongly connect component coupling at each step is the same as in the proof of Theorem 1, but we must simply account for the difference in definitions of $m' = mc$ and $\beta_c = \beta$ because the probabilities are a function of $c$ steps now rather than 1 step before. So we can simply plug these substitutions into our equations from Theorem 1 to show that $\frac{m}{c} \geq \frac{1}{\beta_c^g} log(1/\epsilon)$. The result is then a simple multiplication of the bound from Theorem 1 by $c$ given the new definition of $\beta_c$.

## A.4  Proof of Lemma 1

**Lemma 1** *(Conditioning on Subsets of State Variables): Consider the Markov chains induced by two policies drawn from different policy classes. The first policy is $\pi(\cdot|x)$, which takes $n(t)$ states variables $x = s^1, ..., s^{n(t)}$ as input at time $t$, has $\ell$ maximal strongly connected components with a maximum size of $g$ state variables and a minimum of $\beta$ common probability mass between any two state configurations for any state variable. The second policy $\pi'(\cdot|x')$, only takes some arbitrary subset of $n'(t) \leq n(t)$ state variables as input at any moment in time $t$ such that $x' = s^1, ..., s^{n'(t)} \subseteq s^1, ..., s^{n(t)}$ has $\ell'$ maximal strongly connected components with a maximum size of $g'$ state variables and a minimum of $\beta'$ common probability mass between any two state configurations for any state variable. The mixing times in the policy classes of $\pi$ and $\pi'$ can then be upper bounded such that the bound on $\pi'$ is tighter:*

$$t_{ret}^{\pi'}(\epsilon) \in t_{mix}^{\pi'}(\epsilon) \in \mathcal{O}\left(\frac{1}{\beta'^{g'}}log(1/\epsilon)\right) \in \mathcal{O}\left(\frac{1}{\beta^{g}}log(1/\epsilon)\right)$$

The policy class over $\pi'$ removes edges in the DBN that were previously present in the class over $\pi$ and cannot add any. If these edges were within a pre-existing maximal strongly connected component, that component may be split into multiple smaller strongly connected components, otherwise the number of strongly connected components stays the same. This implies that $\ell' \geq \ell$. If removing edges resulted in splitting the largest strongly connected component, we can say that $g' < g$ and $g = g'$ otherwise. This implies that $g \geq g'$.

To understand the relation between $\beta$ and $\beta'$ we must make our notation a bit more detailed than what was presented in the main text. $\beta_i$ can then be stated as follows while making the parameterization of $\pi$ explicit:

$$\beta_i = \min_{\mathcal{X}(t) \times \Delta(\mathcal{A})} \left[ \min_{s_1, s_2 \in \mathcal{S}} \left( \min\left\{ \sum_{a_1 \in \mathcal{A}} \pi(a_1|x_1)T(s^i|s_1, a_1), \sum_{a_2 \in \mathcal{A}} \pi(a_2|x_2)T(s^i|s_2, a_2) \right\} \right) \right] \tag{8}$$

Likewise, $\beta_i'$ can be defined in a similar fashion:

$$\beta_i' = \min_{\mathcal{X}'(t) \times \Delta(\mathcal{A})} \left[ \min_{s_1, s_2 \in \mathcal{S}} \left( \min\left\{ \sum_{a_1 \in \mathcal{A}} \pi'(a_1|x_1')T(s^i|s_1, a_1), \sum_{a_2 \in \mathcal{A}} \pi'(a_2|x_2')T(s^i|s_2, a_2) \right\} \right) \right] \tag{9}$$

where $\mathcal{X}(t) = \mathcal{S}^1 \times ... \times \mathcal{S}^{n(t)}$ and $\mathcal{X}'(t) = \mathcal{S}^1 \times ... \times \mathcal{S}^{n'(t)}$ such that $\mathcal{X}'(t) \subseteq \mathcal{X}(t)$ for all $t$ while $x_j$ and $x_j'$ are analogously obtained from the full state $s_j$ for $j \in \{1, 2\}$. The main difference between these formulations is then minimizing over $\mathcal{X}(t) \times \Delta(\mathcal{A})$ in the case of $\beta_i$ and $\mathcal{X}'(t) \times \Delta(\mathcal{A})$ in the case of $\beta_i'$. By rule of Cartesian products over subsets, we then know that $\mathcal{X}'(t) \times \Delta(\mathcal{A}) \subseteq \mathcal{X}(t) \times \Delta(\mathcal{A})$ and because the minimization over a subset must yield a result larger or the same size as a minimization of the full set, $\beta_i \leq \beta_i'$ for all state variables $i$. This also implies that $\beta \leq \beta'$ as it also must hold for the smallest value over state variables.

Bringing it all together, $0 \leq \beta \leq \beta' \leq 1$, and $g \geq g' \geq 1$, so $1/\beta'^{g'} \leq 1/\beta^{g}$. So, when combined with the results from Theorem 1, this result has been proven.

## A.5  Proof of Theorem 2

**Theorem 2** *(Limiting Mixing Times with Context Length): If the Markov chain induced by a policy conditioned on a finite interaction history window $\pi(\cdot|h^{(k)})$ with a context length of $k$ has $\ell_k$ maximal strongly connected components in $\mathcal{D}^\pi$ with a maximum size of $g_k$ variables and a minimum of $\beta_k$*

*common probability mass between any two state configurations, the mixing times $t_{ret}^{\pi}(\epsilon)$ and $t_{mix}^{\pi}(\epsilon)$ can be bounded for any $k' \geq k$:*

$$t_{\mathrm{ret}}^{\pi}(\epsilon) \in t_{\mathrm{mix}}^{\pi}(\epsilon) \in \mathcal{O}\left(\frac{1}{\beta_k^{g_k}} log(1/\epsilon)\right)$$

$$\in \mathcal{O}\left(\frac{1}{\beta_{k'}^{g_{k'}}} log(1/\epsilon)\right)$$

*where the dependence on the context length implies that $0 \leq \beta_{k'} \leq \beta_k \leq 1$ and $g_{k'} \geq g_k \geq 1$.*

**Proof Sketch:** Lemma 1 considers the mixing time relationship of policy classes conditioned on subsets of the state variables that other policy classes are conditioned on. Our proof includes the following high-level steps by applying our notation from Section 3 in which $k' \geq k$ for all $t$ to the results of Theorem 1 and Lemma 1:

1. We consider the causal parent state variables of each observation, action, and reward to conclude that $Par(h^{(k)}) \subseteq Par(h^{(k')}) \subseteq \{1,..,n\}$, which implies that $n \geq n_{k'}(t) \geq n_k(t)$.

2. Through Lemma 1 we show that by rule of Cartesian products over subsets that $0 \leq \beta_{k'} \leq \beta_k \leq 1$.

3. Through Lemma 1 we also demonstrate that $g_{k'} \geq g_k \geq 1$ because causal connections in $\mathcal{D}^{\pi}$ are only added and not removed when the context length is increased.

4. This then implies that $1/\beta_{k'}^{g_{k'}} \geq 1/\beta_k^{g_k}$, which is sufficient to prove Theorem 2 using Theorem 1 because $\epsilon$ is independent of $k$.

**Additional Details:** As mentioned in the section overview, we assume that observation $o$, action $a$, and reward $r$ are caused by of a subset of the state variables such that $Par(o) \subseteq \{1,..,n\}$, $Par(a) \subseteq \{1,..,n\}$, and $Par(r) \subseteq \{1,..,n\}$. As a result, any interaction history window $h^{(k)}$ of size $k$ is also caused by a subset of the state variables such that $Par(h^{(k)}) \subseteq \{1,..,n\}$. This is always true in practice as this includes as a corner case the situation where the interaction history is dependent on all variables. However, our bound is not meaningful in domains where $Par(h^{(k)}) = \{1,..,n\}$ for all values of $k$ because then it never gets tighter as $k$ gets smaller. In general, a finite context length is reflective of a maximum $n_k(t)$ sized subset of the state variables at a time $t$ where $n \geq n_{k'}(t) \geq n_k(t)$ if context length $k' \geq k$ for all $t$. This is because $Par(h^{(k)}) \subseteq Par(h^{(k')}) \subseteq \{1,..,n\}$. Theorem 2 then follows directly from the combined results of Lemma 1 and Theorem 1.

Specifically, $0 \leq \beta \leq \beta_{k'} \leq \beta_k \leq 1$ because they are the result of minimizing the same function over progressively smaller subsets as in Lemma 1. Moreover, $g \geq g_{k'} \geq g_k \geq 1$ because causal connections in the DBN are only removed as in Lemma 1. This then implies that $1/\beta^g \geq 1/\beta_{k'}^{g_{k'}} \geq 1/\beta_k^{g_k}$, which demonstrates the relative tightness of the bounds. Finally, Theorem 1 does not depend on the parameterization of the policy, so it can equally be applied to policies of history length $k$ to bound the mixing time.

## B  Additional Details for Experiments

**Section Overview:** Here we elaborate on experimental details that we did not have room to include in the main text. In each subsection, we fill in missing details from Sections 2, 3, and 4 respectively.

**Compute Infrastructure:** Our experiments were deployed on a cluster of Intel x86 machines. Each of our toy experiments and tabular experiments were run on a single CPU. Meanwhile, the online function approximation and Decision Transformers experiments were each run with one V100 GPU.

**Software Libraries:** Simple tabular models were coded from scratch primarily using Numpy [36], which is publicly available following a BSD license. Neural network models were developed using Pytorch [37], which is publicly available following a modified BSD license.

**Hyperparameter Tuning Protocol:** Our models train to convergence in all experiments. For our simple RGB world experiments, we did a search over Adam learning rates with gradient clipping

until finding one that consistently converged at every context length. For our Decision Transformers experiments, we followed the code from the original paper by Chen et al. [19], which included gradient clipping, Adam optimization, and a learning rate schedule that consists of linear warm-up following by cosine learning rate decay. These details were copied from the original paper [19] and we also did a grid search over the initial learning rate.

### B.1 Toy Examples (Section 2)

**Mixing Time Calculation:** Following Riemer et al. [14], we compute a relaxed version of the $\epsilon$-return mixing time that is averaged over start states to be more reflective of mixing times encountered rather than the worst case. This is computed by first rolling out the policy for 100,000 steps in the environment to approximate the reward rate from the first state in our list. This state is arbitrarily chosen such that all variables are set to index 0. Then for each environment state we rollout the policy for 10,000 steps, recording the final step at which the environment is not within $\epsilon$ of this reward rate. The reported mixing times in the main text are an average across all possible starting states. For the smaller irrelevant variables example, we also compute the mixing time for each starting state 5 times and take the average over trials in order to lower the variance of our estimates.

**Irrelevant Variables Example Details:** For the irrelevant variables example, there are 9 total states across variables $x$ and $y$ and only 3 total states when ignoring $y$. As a result, there are $2^9 = 512$ deterministic policies across both variables and $2^3 = 8$ deterministic policies over the relevant variable. $0.474$ is the highest reward rate estimate for any policy, so if the reward rate has not mixed to a precision of $\epsilon = 0.05$, there must be significant bias and greater than $10\%$ relative error in the reward rate estimation regardless of the policy.

**Independent Subtasks Example Details:** For the independent subtasks example, there are 18 total states across all variables $x$, $y$, and $z$ and only 6 total states when ignoring the irrelevant variable for the subtask. As a result, there are $2^{18} = 262,144$ deterministic policies across all variables and $2^6 = 64$ deterministic policies over the contextually relevant variables. $0.474$ is again the highest reward rate estimate for any policy, so if the reward rate has not mixed to a precision of $\epsilon = 0.05$, there must be significant bias and greater than $10\%$ relative error in the reward rate estimation regardless of the policy. As the search space over deterministic policies considering all variables is very large for this domain, we randomly sort the policies and only evaluate policies for 24 hours on a single cpu. 1,597 total policies were considered. Notice that this means the numbers reported in the main text are conservative as less than an arbitrary $1\%$ of the search space was actually explored to find these values.

### B.2 Online Learning in a POMDP (Section 3)

**Statistical Significance:** The lightly shaded regions around each data point in Figures 4, 5, and 7 represent the 95% confidence interval of the average mixing time of that point given the number of data points found at that particular binning with respect to the reward rate across runs and seeds. This is computed as $\pm 1.96\sigma/\sqrt{n}$ where $\sigma$ is the standard deviation and $n$ is the number of data points at a particular binning of the reward rate.

**Experimental Protocol:** For each setting considered, we conducted a version of the experiment with 100 different seeds ranging from 0 to 99. Our tabular experiments were run for 1 billion steps (corresponding to about 7 days for the highest context length $k = 10$). Meanwhile, our function

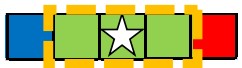
(a) Online Learning: RGB World

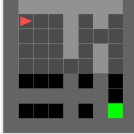
(b) Offline Learning: Minigrid Crossing

Figure 6: **Domains.** Figure 6a depicts the environment used for online learning with left and right actions. The reward is $+1$ if the agent is on the blue square and $-1$ on the red square. If either square is reached, the agent returns to the center while the red and blue squares are randomly assigned to the edges. Figure 6b depicts a random episodic configuration of the environment used for offline learning. Observations are denoted by yellow dotted lines or light shading.

approximation experiments were run for 6 hours each, corresponding to about 1 million steps for the Transformer models, 6 million steps for the MLP models, and 600 thousand steps for the LSTM models. We set the number of green squares to 3, which implies that the $k = 1$ policies cannot achieve the optimal policy, but all $k \geq 2$ can. Based on the results of preliminary testing, the SGD learning rate was set to 0.01 in our tabular experiments and the Adam learning rate was set to 0.000001 in our function approximation experiments. Online Q-learning with an exploration rate of 0.05 and a discount factor of $\gamma = 0.99$ was used in all experiments. During our function approximation experiments, the target network was updated every 10,000 steps following best practices [38].

**Neural Network Architecture Details:** For our MLP models, the history representation is flattened at each step and sent to a two layer MLP model with 724 hidden units in each layer and a linear layer deriving an output for each action's value. ReLU activations are used at each hidden layer. Our LSTM model is sent a sequence of 5 dimensional inputs for each step where 3 dimensions represent the observation, one dimension represents the action, and one dimension represents the reward. The current observation is then sent to the LSTM with an action of $-1$ and reward of 0 in order to preserve the dimensionality. The LSTM, implemented with the Pytorch [37] LSTM library, is unidirectional and has two hidden layers of size 256 followed by a linear layer to produce a value for each action. The input sent to the Transformer model has 6 dimensions at each step, adding to the LSTM input an encoding of the ordering of the input ranging from 1 to $k$. Our model is implemented using the Pytorch [37] Transformer library leveraging a hidden dimension of 256, one encoder layer and one decoder layer. The output of the Transformer is once again sent to a linear layer producing a value for each action. The number of heads was set to 8 based on preliminary experiments.

**Mixing Time Calculation:** We adapt our mixing time estimation approach from the last section as we cannot easily generate the full set of possible histories corresponding to each state. Instead we rollout a single chain from the current state for 10,000 steps using the greedy exploitation policy. At each step along the way, we calculate the estimated reward rate and report the mixing time as the last time where the estimated reward rate is not within $\epsilon = 0.01$ of the full estimate over all of the steps. This is similar to the approximation used by Riemer et al. [14] for larger scale domains.

**Mixing Times Across Context Lengths:** Our results in the main text only plot the mixing time for the context length $k = 5$. We provide more comprehensive results across values of $k$ in Figure 7. These additional settings follow a similar overall trend. Reward rates are binned by increments of 0.0333.

**Parameter Scaling Experiments:** We varied the number of hidden units in each architecture in order to test the effect of model size on encountered mixing times over 100 random seeds and the same learning configuration.

- **MLP Scaling.** For the MLP model we considered: 256 hidden units resulting in 134,402 total parameters (at $k = 2$), 404 hidden units resulting in 331,686 total parameters (at $k = 2$), 724 hidden units resulting in 1,057,766 total parameters (at $k = 2$), 1280 hidden units resulting in 3,293,442 total parameters (at $k = 2$), and 2250 hidden units resulting in 10,154,252 total parameters (at $k = 2$). While the MLP model does grow with the context length, most parameters are not in the first layer and this only results in 8.7% relative parameter growth from a context length of 1 to 10. We considered both $k = 2$ and $k = 10$ for each number of hidden units and found no statistically significant effect varying the number of parameters while keeping $k$ fixed.

- **LSTM Scaling.** For the LSTM model we considered: 92 hidden units resulting in 137,634 total parameters, 143 hidden units resulting in 330,618 total parameters, 256 hidden units resulting in 1,054,722 total parameters, 454 hidden units resulting in 3,308,754 total parameters, and 792 hidden units resulting in 10,055,234 total parameters. We again considered both $k = 2$ and $k = 10$ for each number of hidden units and found no statistically significant effect varying the number of parameters while keeping $k$ fixed.

- **Transformer Scaling.** For the Transformer model we considered: 88 hidden units resulting in 127,338 total parameters, 136 hidden units resulting in 301,242 total parameters, 256 hidden units resulting in 1,058,562 total parameters, 448 hidden units resulting in 3,228,738 total parameters, and 792 hidden units resulting in 10,067,114 total parameters. We plot the effect on the average mixing time encountered during learning in Figure 8. We use this aggregate metric in this case only to simplify our analysis given that we are also varying the number of parameters, but it is definitely preferable to plot mixing times as a function

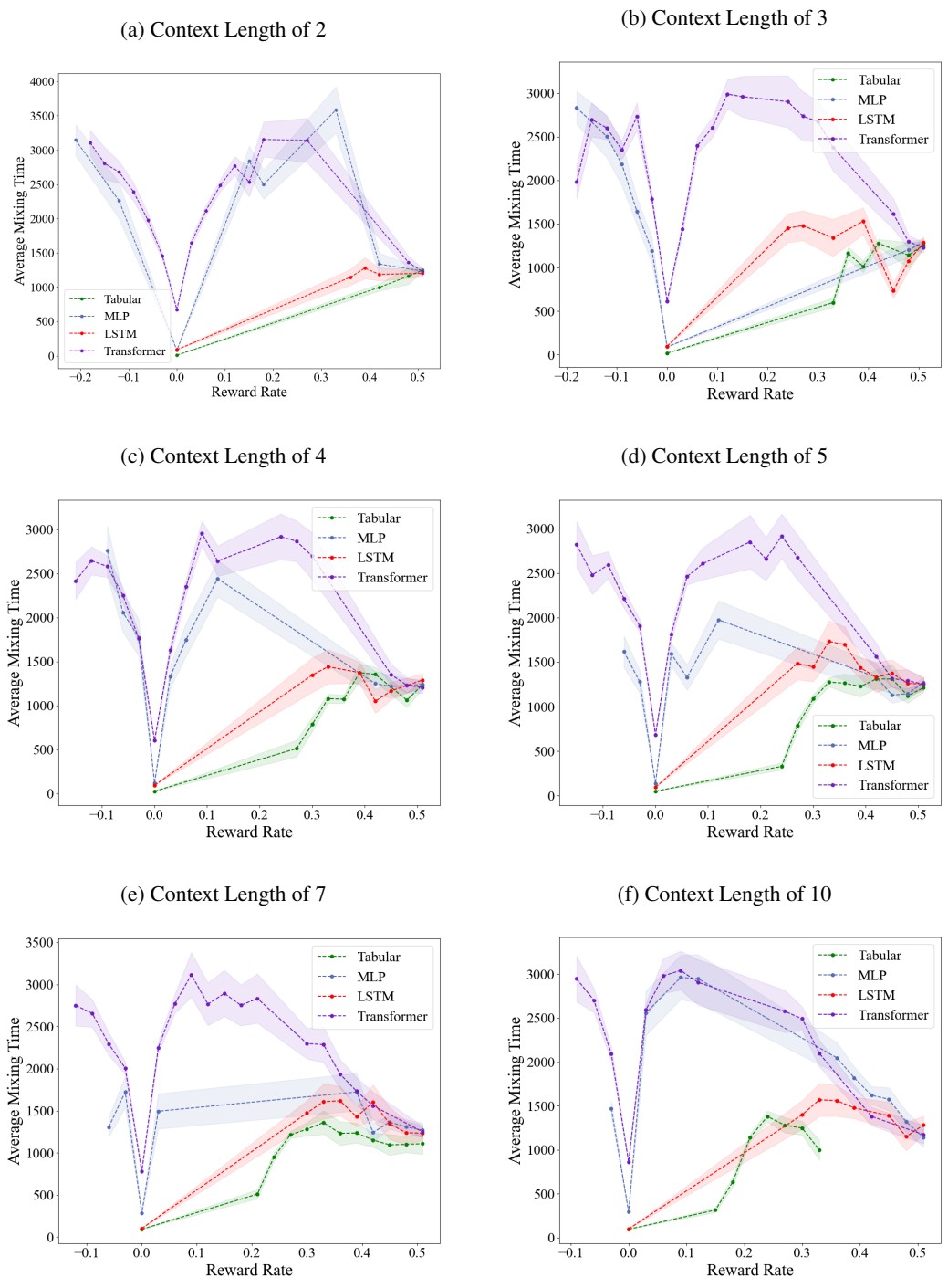

Figure 7: **Policy Architecture vs. Encountered Mixing Times.** We plot the average mixing time for each choice of policy architecture between tabular, MLP, LSTM, and Transformer models with averages binned by the reward rate of the policy.

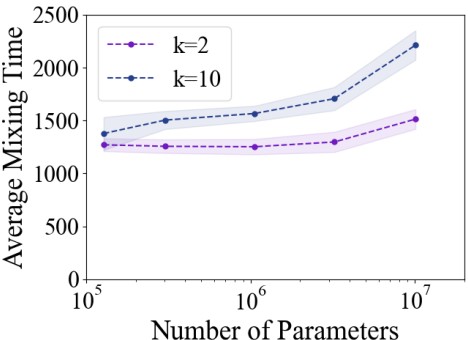

Figure 8: **Transformers: Average Encountered Mixing Time vs. Number of Parameters.** We plot the average encountered mixing time during learning for Transformer models trained at a number of different model sizes regulated by the number of hidden units. Averages are taken over 100 random seeds and we also provide 95% confidence intervals.

of the reward rate as we have primarily done throughout this paper. Because mixing times based on the $\epsilon$-return mixing time depend strongly on the reward rate, aggregate metrics like this largely wash out the effect of other factors. In light of this, the nearly 2x increase in the average encountered mixing time over the course of the entire learning period between $k = 2$ and $k = 10$ for Transformers with over 10M parameters is quite staggering and showcases the significant impact of the analysis in Theorem 2 even in this simple domain.

**Transformer Attention Map Analysis:** To test the hypothesis of the model size increase being correlated with paying more attention to the full context, we reran our experiments for 136 hidden units and 448 hidden units (10x more parameters) at $k = 10$ while keeping track of the attention maps computed in the decoder with respect to the interaction history at past time-steps. During the mixing time evaluation phase, we averaged the attention maps computed at each step and then averaged over evaluations weighted by the number of steps between evaluations. The average attention weights for the 136 hidden unit model were: $t - 9 : 0.048$, $t - 8 : 0.052$, $t - 7 : 0.061$, $t - 6 : 0.078$, $t - 5 : 0.099$, $t - 4 : 0.123$, $t - 3 : 0.148$, $t - 2 : 0.179$, and $t - 1 : 0.212$. In contrast, the average attention weights for the 448 hidden unit model were: $t - 9 : 0.089$, $t - 8 : 0.080$, $t - 7 : 0.078$, $t - 6 : 0.086$, $t - 5 : 0.102$, $t - 4 : 0.119$, $t - 3 : 0.133$, $t - 2 : 0.149$, and $t - 1 : 0.163$. The average attention weight goes up for all time-steps in the interaction history window greater than 4 steps in the past. Indeed, the entropy associated with the average distribution grows over 4% from 2.99 bits for the 136 hidden unit model to 3.12 bits for the 448 hidden unit model.

### B.3  Offline Learning with Decision Transformers (Section 4)

**Environment Details:** Documentation for the *SimpleCrossingS9N1-v0* environment that we used for our experiments can be found at `https://minigrid.farama.org/environments/minigrid/CrossingEnv/`, which is publicly available following the Apache 2.0 license.

**Decision Transformer Details:** We model our code and architecture following the code provided for the Atari experiments of the original Decision Transformers paper [19], which was released under an MIT license. Our architecture and optimization follows all of the details from their paper with the only exception being the use of a convolutional architecture that his been found more suited to the Minigrid domain where observations are $3 \times 7 \times 7$ dimensional [30; 31]. There are three convolutional layers with filter sizes of 16, 32, and 64 respectively. $2 \times 2$ max pooling layers and ReLU activations follow each convolutional layer. The result is a 64 dimensional embedding (as opposed to 128 for Atari) due to the smaller input size. Following Chen et al. [19], we used a batch size of 128, 6 layers, 8 attention heads, 5 epochs of training, GeLU nonlinearities within the Transformer, $512 * 20$ warm-up tokens, $2 * 500000 * k$ final tokens, and a dropout rate of 0.1. We also used the same Adam optimization with a learning rate of 0.001, $\beta = (0.9, 0.95)$, a gradnorm clip range of 1.0, and 0.1 weight decay. It is important to note that a transformation is applied to the rewards before it is sent to

the Transformer model referred to as the *return to go* for policy or trial $i$ namely $g_{i,t} := \sum_{\tau=t}^{T_i} r_{i,\tau}$ which is sent along with the current time step $t$ instead. The advantage of this representation is the ability to prompt the model for high returns to elicit high performing behaviors. We can consider this as a special case of the paradigm in which models take as input some transformation of the interaction history up to a bounded context length.

**Random Behavior Policies:** Our random behavior policies take in the current observation as input and process it with the same convolutional encoder as the Decision Transformer model. This representation is simply processed by a linear layer, mapping it to a value for each action. The policy then selects the argmax action. The maximum length of any episode is 325, the minimum length is 14, and the average length is 324.4 (as most random policy episodes do not arrive at the goal state).

**Learning Behavior Policies:** Our learning behavior policies are implemented with the REINFORCE algorithm [32]. It is quite common in the offline RL literature to use data drawn from a replay buffer during learning. However, the correspondence of these to actual policies used can get complicated with an off-policy algorithms such as DQN or Rainbow. Meanwhile, REINFORCE only performs updates after each episode and can ensure that the actions in each episode are drawn entirely from on-policy sampling. We opted for this approach due to its simplicity and to control for other potential complicating factors in the learning process such as policies that change or randomly explore in the middle of an episode. The learning rate of REINFORCE is set to 0.1 with SGD optimization, the entropy regularization coefficient is set to 0.001, and a discount factor is set to $\gamma = 0.99$. The maximum length of any episode is 325, the minimum length is 14, and the average length is 240.5. We observe that by 10,000 episodes of training the policy is consistently solving the task, but we consider just the first 1,000 episodes to make the comparison more precise with the random behavior policy setting and to increase the variety in the behavior distribution (by putting some emphasis on early learning too).

**Mixing Time Calculation:** We aim at a similar evaluation of the mixing time as the previous section. However, we also take advantage of the episodic structure of this problem to promote computational efficiency. The policy is rolled out for 1000 episodes to compute the estimated reward rate per step and reward rate per episode. Then we consider 1000 random orderings of these episodes keeping track of both the final step not within $\epsilon = 0.01$ precision of the estimated reward rate per step and the final episode not within $\epsilon = 0.01$ precision of the estimated reward rate per episode. We report the average mixing time over these 1000 random orderings to again highlight mixing times experienced in practice rather than worst case mixing times.

**Evaluation of Decision Transformer Models:** In Figure 9 we take a deeper look at the generalization ability of models trained with decision transformers as a function of the context length $k$. We now elaborate a bit on our discourse from the main text. One may wonder if it is necessary for Decision Transformers to achieve near optimal training accuracy in order to achieve good downstream performance. To answer this question, we consider a Decision Transformer that we trained on random behavior policies (seed 0) that achieved optimal training accuracy at $k = 25$. This particular setting is interesting because this is the smallest $k$ found to be sufficient and because the deterministic behavior policies allow us to identify the best possible training accuracy given the data (100%). We then evaluate this model across a variety of return to go prompts stepping by 0.01 from 0 to 1.0 and report the average performance of these models across 1,000 episodes. In Figure 9 we plot the reward rate of these models ranked according to performance (along the x axis) as well as each behavior policy used to generate the data evaluated in the same way. We plot our results in Figure 9 which validate our hypothesis about the importance of fitting the training data. Decision Transformer with $k = 25$ can achieve performance as good as any behavior policy used to generate the data. Meanwhile, models with smaller $k$ fail to achieve the same performance and models with larger $k$ seem to overfit on a dataset of this size.

# C   Why Average Reward RL?

**Issues with Discounting:** As typically implemented, the popular discounted reward setting of RL does not correspond to the maximization of any objective function over a set of policies [39] and the policy gradient is not the gradient of any function [40]. Moreover, these fundamental issues do not resolve as the discount factor approaches 1 [39] and discounting does not influence the ordering of policies, suggesting it likely has no role to play in the definition of the control problem [21]. In

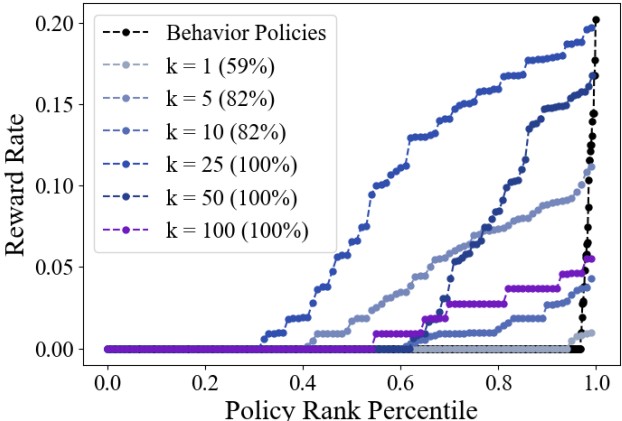

Figure 9: **Decision Transformers Context Length vs. Reward During Evaluation.** We highlight learning results for Decision Transformers in the crossing environment with data generated from random behavior policies with a context length $k$ of 1. This figure details the evaluation performance when prompting each policy with 100 return to go values stepping from 0 to 1.0. This is also compared to the actual distribution of performance for the behavior policies that generated the data. Each line color represents a different context length $k$ for the Decision Transformer model with the training accuracy rounded to the nearest percent included in parenthesis.

contrast, the average reward per step objective, is well-suited for analysis of partially observable [1], larger scale [14], and continual RL problems [21; 27; 41]. We thus adopt this formulation in our paper to enable more rigorous analysis of agents.

**Evaluation vs. Optimization:** It is important to note that we only use the average reward perspective to evaluate an agent's behavior and place no restriction on discounting being used within the agent's learning process as it is in our experiments. So, to argue that the discounted setting would be more relevant for our goals, is essentially to propose that what practitioners actually care about is the discounted return rather than the undiscounted return or average reward of policies. As highlighted by Schwartz [42] it is quite rare for actual evaluation results reported in papers to consider discount factors. Meanwhile, optimizing for the average reward actually also inherently optimizes for the undiscounted return [42]. That said, even if we did really care about the discounted return, this is a corner case subsumed within the formulation of Theorem 2. This is because the discount factor or time-step number can be considered to be a potentially unobserved element of the state space that is used to modulate the function that produces immediate rewards. As such, our choice of the average reward per step formulation does not at all limit the generality of our conclusions in Theorem 2. In fact, it helps ensure that the result is as general as possible.

**Infinite vs. Finite Horizons:** Bojun [43] proves that every finite horizon task has a unique steady-state distribution under any policy. This means that Assumption 1 and Corollary 1 are guaranteed for the case of episodic tasks, which is subsumed within our framework. For episodic tasks, the mixing time is related to the number of episodes a policy must be rolled out for before we get a reliable measure of performance. Indeed, we consider episodic tasks within our experiments using the Minigrid Crossing environment in Section 4.

## D  New Architectures and Algorithms

Theorem 2 shows that what leads to potentially high mixing times is when models leverage a monolithic representation that is highly sensitive to a large part of the interaction history at all times. This is particularly descriptive of how vanilla transformers work, but there are multiple already existing research directions that seem well suited to scaling to high context lengths while providing less history sensitivity at each step, which we will now outline.

**Hierarchical RL:** In hierarchical RL frameworks such as options [44] when applied to neural networks [45] with potentially many levels of abstraction [46] and/or complex weight sharing

[47; 48; 49; 50], it should be possible in domains with temporal coherence to approximate a policy with a longer context length by multiple sub-policies with a smaller context length. This is particularly relevant for approaches to option learning that learn an independent lower dimensional representation space for each option as in [31], or associate options with subsets of the state space [51; 52; 53; 54].

**Hybrid Transformer Working Memory Architectures:** Recent work has aimed to improve the effective context length of transformers by augmenting them with some kind of bounded working memory component [55; 56; 57]. While these papers are typically solely motivated by computational efficiency, they effectively serve the role of extending Transformers to longer contexts while making them less sensitive to experiences in this context that are not reflected in the memory. This effectively brings Transformers closer to some of the incremental design patterns of RNNs, which our experiments indicate experience lower mixing times. Approaches that have less sensitivity to the input space are also motivated by the problem of attention dilution [58; 59; 60; 61], which is typically addressed with retrieval augmented approaches that only focus on a subset of the interaction history at a time [62; 63; 64] or a generated small description [65].

**Tracking Policies:** If we aim to learn a non-stationary tracking policy solution concept as argued by [66] we could potentially represent a stationary policy over a longer context length with a non-stationary set of small context length policies. In effect, this becomes similar to what is achieved by the hierarchical RL policies described above. One additional subtlety to highlight in this case is that this solution is also limited by the mixing properties of the tracking policy parameters, so it would also be necessary to tune this approach to move through parameter space as fast as possible.

**The Role of Replay:** While not related to sensitivity of the policy to its input, it is worth noting that replay based learning algorithms have been theoretically shown to lead to better sample efficiency of learning with respect to the mixing time [67; 16]. It will be an interesting direction for future work to explore the extent to which these learning benefits can be combined with the benefits of a particular choice of policy class aimed at lowering mixing times with a smaller context length. Moreover, considering the importance of our analysis to continual learning settings, it will be interesting to see if similar mixing time benefits could be established for approaches that combine replay with meta-learning [68], or approximate replay through the use of strong generative models [69].

