# OpenReview forum: "Balancing Context Length and Mixing Times for Reinforcement Learning at Scale"
_NeurIPS.cc/2024/Conference — NeurIPS 2024 poster_

### Official Review · Reviewer_aEyh · 2024-07-11

**Soundness:** 2
**Presentation:** 4
**Contribution:** 3
**Rating:** 6
**Confidence:** 3

**Summary:**

This paper studies the interaction between policies that operate given some context and the mixing time of the policies. It provides a novel connection between the context length and the mixing time, indicating that increased context length might lead to increased mixing times and thus, need for longer, more expensive evaluation. The paper provides a theoretical connection in form of two theorem statements, provides toy examples and ultimately demonstrates some of the findings in an experimental section.

**Strengths:**

Problem Statement
* Understanding the complexities of context-based reinforcement learning is an important and interesting problem

Clarity
* The paper was a pleasure to read to due its high quality of writing
* The paper is very well structured and the story thread is clear, the reader is walked through various sections with simple examples that help foster understanding

Related Work
* A good amount of related work is spread throughout the paper and given the high quality of writing I do not believe a separate section for related work is needed.

Theoretical results
* While the theoretical results are not ground breaking in their complexity, they offer a nice novel spin on some known results and provide a new angle of looking at context lengths which I think is great.

Experiments
* Simple toy examples verify some of the findings proposed by the theory.

**Weaknesses:**

Clarity
* Some of the Figures in the experimental section took me a bit to understand, specifically how the reward rates were correlated with the mixing time in Figures 3 and 4.
Theory
* The provided bounds are upper bounds and just because they are slightly tighter for lower context lengths does not mean that any algorithm will follow the trends of the these bounds. It could very well be that any context length complexity is below the complexity of the smallest bound. However, I think this is a good first stab at establishing a theoretical trend that might carry into practice.

Experiments
* This is I think the weakest part of the submission as I do not find all the experiments conclusive. I think there might be other confounding factors at work that have not been determined yet. Some of my concerns might be alleviated when my open questions.
  * In both Figure 3 and 4, it seems that policies with high reward rates converge to the same mixing time independent of the context. This questions whether the claim of correlation between the two is correct.
  * In Figure 4, there seems to be a clear difference between different models but it is unclear to me whether this is a separate issue or whether it can actually be attributed to context length since the MLP seems to have significantly lower mixing time than the Transformer but they use the same context lengths.
  * In Figure 9, the increased context length might not only raise the mixing time but also introduces a vast amount of additional parameters that need to be fit. This makes it unclear if the mixing time or other potential causes lead to the observed behavior.

Overall, I think this paper provides a novel idea and some new insights into analysing context lengths and even though not all the experiments are conclusive, I am leaning towards acceptance since I believe that the community will benefit from this work being available to them. I believe this is an interesting direction that should be further explored. I am arguing for a weak acceptance for now but am willing to raise my score if my open questions can be addressed.

**Questions:**

Q1: In Figure 3, why does k=10 end at reward rate 0? Does it never achieve higher reward? If so, can you explain why?

Q2: Do you have the fitting curves for the transformer models? Did they train to convergence? How did you tune the hyperparameters? Transformers can be a little tricky to get right and I’m curious what efforts were made to ensure proper function fitting.

Q3: In Figure 4, do you have any results on the state-occupancy distributions of the policies at different reward rates? Or the similarity of these policies? It seems that the random policies which mix well and achieve 0 reward have similar mixing times and the optimal policy which possibly always executes the same short trajectory has similar mixing times. I am currently not 100% convinced that this is a function of context length.

Q4: In Figure 4, do you have an idea why the transformer has much larger mixing times than the MLP? I’m assuming both see the same context length.

Q5: I might be misunderstanding Figure 5, but why would we expect (no matter the context length) a model to be able to distinguish between 1000 different random policies unless it can hold the length of the trajectory in memory?

Q6: It seems to be that there is a difference between the context required to express a policy and the context available to the model. How can this be modeled within the framework? The optimal policy of the MDP described in 6a clearly does not require a very long context. Why would the model with large context not simply learn to ignore any token farther away than like 3 steps (given enough data)?

**Limitations:**

The limitations of the theoretical assumptions have been discussed. However, I believe a more critical treatment of the experimental results would benefit the community to discern where we can be certain about the correctness of the correlation.

---

> ### Author Rebuttal · Authors · 2024-08-06
>
> Thank you for your comprehensive review of our paper. We really appreciate your kind words about the importance of our problem statement, the clarity of our writing, the quality of our theoretical results, and the value of our work to the research community. We have tried to address each point of confusion that you mentioned below:
>
> **Q6:** While you asked this question last, we wanted to address this one at the onset as it is related to a few other points of confusion that you mentioned. We really like the way that you put “there is a difference between the context required to express a policy and the context available to the model.” We will update Sections 2 and 3 to emphasize this point. For example, let’s consider the irrelevant variable example from Figure 1 where it is only necessary for the policy to leverage $x$ and not $y$. Because the optimal policy only depends on $x$, this policy is included within the set of policies that only depend on $x$. However, the optimal policy is also included within the set of policies that take both $x$ and $y$ as input. Even though the functional form of this policy depends on both $x$ and $y$, the actions of this particular policy are only conditionally dependent on $x$. So regardless of the functional form used to search for the optimal policy, the optimal policy is the same in both cases. It receives the same reward rate and encounters the same mixing time. This is why we focused on $\epsilon$-optimal policies that depend on both $x$ and $y$, making the point that there are very good policies with higher mixing times than any policy that only depends on $x$. So yes, any policy that receives input that is a sufficient statistic of the environment state and trains for long enough with a principled learning algorithm will arrive at the same optimal policy. The differences is not at this theoretical asymptotic limit, but rather at the points encountered along the way. In real-world problems of sufficient complexity, arriving at the truly optimal policy is generally considered infeasible.
>
> **Convergence in Figure 3 and 4:** As described above, the optimal reward rate is associated with the same optimal policy regardless of the context length and model class. As noted in the caption of Figure 3, policies with context length $k=1$ do not have sufficient context to achieve the optimal policy. However, every policy class that takes in a greater context length, does eventually achieve the same optimal policy by the end of training. What is more interesting is the mixing times associated with policies experienced along the way for each model class.
>
> **Transformers vs. MLP (Q4):** To clarify all models in Figure 4 use the same context length and Figure 7 of the appendix includes a chart for each of the other context lengths. Keep in mind that we were careful to control for the number of parameters so that the architecture is the main source of variation. The comparison with MLPs is not straightforward and we should note that they do have the highest mixing times relative to Transformers when looked at across different context lengths, experiencing similar mixing times to Transformers i.e. for $k=10$. We hypothesize that these results indicate that MLPs are on average a bit less sensitive to the full context than Transformers when considered over the full course of learning.
>
> **Number of Parameters:** We are not sure that we were fully able to follow this comment. Figure 9 is from experiments using the Decision Transformer architecture, which does not introduce more parameters when dealing with longer contexts due to parameter sharing within the sequence model.
>
> **Q1:** Thank you for pointing this out. This was a simple artifact of our plotting library which filtered out the points at reward rates of 0.4 and 0.5 for $k=10$ just because there was a bit less data (because of slower learning). When we take out this filter, we predictably see that the mixing time is above the other context lengths at a reward rate of 0.4 and, as expected, converges to the same optimal policy mixing time as the other context lengths. We are really sorry about this mistake and will update the final draft to address this omission.
>
> **Q2:** Yes. Our models train to convergence in all experiments. For our simple RGB world experiments, we did a search over Adam learning rates with gradient clipping until finding one that consistently converged at every context length. For our Decision Transformers experiments, we followed the code from the original paper, which included gradient clipping, Adam optimization, and a learning rate schedule that consists of linear warmup following by cosine learning rate decay. These details were copied from the original paper and we did a grid search over the initial learning rate. We will add these details in Appendix B.
>
> **Q3:** As explained above, the optimal policy is the same regardless of the function class, so convergence of all models around a reward rate of 0.5 is expected. We hypothesize that when multiple model classes have the same reward rate and same mixing time, they are likely learning the same simple policy i.e. always go right regardless of the state. In general, all policies that do not depend on the state of the environment at all will receive a reward of 0 for this problem and would also have the minimal mixing time bound following Theorem 2. We also really like your idea to look at the state occupancy directly and will do a deeper investigation into what is learned by each model at a reward rate of 0 in the appendix of the final draft.
>
> **Q5:** There is indeed no guarantee that it will always be possible to achieve a training accuracy of 100 percent with limited context. This is actually exactly the point we are trying to make in this figure. Our argument is that models like decision transformers must leverage context lengths far greater than the context lengths used for any behavior policy i.e. $k=1$ in this example.

---

> > ### Comment · Reviewer_aEyh · 2024-08-12
> >
> > Dear authors,
> >
> > thank you for responding to my questions so thoroughly. I also read the reviews by other reviewers. I believe that I have a better understanding of some of the points now. I do maintain that I think this paper introduces novel ideas. I also maintain that the relation between experiments and theory could be stronger as others have pointed out but I believe that this should at this point not be reason for rejection. I think this is a good starting point and I think we should encourage more work in this direction. Thus, I'm raising my score to 6.

---

### Official Review · Reviewer_hXkG · 2024-07-12

**Soundness:** 3
**Presentation:** 3
**Contribution:** 3
**Rating:** 6
**Confidence:** 5

**Summary:**

This work presents theoretical and empirical evidence that increasing the context length informing a policy also increases the amount of time required to evaluate the capability of the policy accurately (in other words for the distribution over states independent of initial state to stabilize). Theorem bounds the mixing time with an inverse relationship between the minimum probability of reaching the same state configuration from two different states, and the number of strongly connected states: demonstrating that with more states with overlapping paths from the policy, the longer the mixing time. Theorem 2 then extends this by incorporating context length, where increasing context length increases the number of strongly connected components. Finally, some experiments are presented which show a general trend in agreement with the theory.

**Strengths:**

# Originality
While this work is based on Kearns and Koller, the authors justify their additions and extend the results far enough to be considered original. In addition the connection to the context length is new and original. The evaluation of foundation models also seems new.

# Quality
The work provides clear theorem statements and in one case a formal statement of the assumptions. Experiments seem valid and do depict a discrepancy in mixing time, as predicted by the theory. Overall the aim of the work is clear and the structure of the work makes sense for evaluating or achieving this aim.

# Clarity
The work is well written and structured which aids reading. Notation is introduced and defined where necessary

# Significance
This work offers a high degree of significance as it demonstrates an important limitation in the evaluation of policies. I believe the potential significance of this work and the subsequent work it may influence is it's biggest strength.  I also agree with the authors that some insight into the reliability of our assessment of foundations models would be useful and timely.

**Weaknesses:**

# Clarity
My biggest concerns of this work are on clarity. It defense of the work - a lot of ground is covered which makes it difficult to structure perfectly. However, in cases throught out terminology is introduced without definition (such as "aperiodic" and "unichain" in Assumption 1 - while I appreciate these are not new terms and I understand what is being said, it still hinders clarity), some notation and concepts are introduced without a clear purpose (for example why bring up coupling time in the middle of DBNs when it doesn't even appear in Theorem 1? If it is a necessary concept for the theorem then this needs to be clear in the theorem. The link between the causal parent state variables and Theorem 2 is similarly unclear and hinders clarity). Finally, the figures, their captions and corresponding examples are not as helpful as they should be. The examples using Figure 6 in Appendix A make no reference to strongly connected components or any of the more technical details in the Theorems. The example which corresponds to Figure 1 is helpful but the figure itself and the corresponding caption lack detail or any clear direction. This could easily be improved as the figure appears to be making the point that the more coordination between state variables the greater the number of strongly connected components. This was a missed opportunity to clarify one of the more opaque concepts in the mathematics. An example where the paper does provide clear explanations comes on lines 263 to 268.

# Quality
I have two concerns on quality
Firstly, the mechanism through which the insight of this work are obtained is not presented. This is due to the absence of proof sketches or any following explanation after the Theorems. A particularly clear example of this is in Theorem 1 where the variable $l$ is not in the bound at all, yes immediately after the statement is made based on the number of strongly connected components (what $l$ quantifies). Another example, is where Lemma 1 is referenced in the Appendix with no explanation or statement of it, when discussing variable subsets. A final weaker example is how $k$ does not appear in Theorem 2 in its own right. While, in this case, it is discussed how $k$ affects $\beta$ and $g$ it would still be useful to get a concise discussion of the exact mechanism of how $k$ actually leads to a larger mixing time. An example where the paper achieves such a type of statement is again on lines 263 to 268, and specifically "When observations are only caused
by a subset of the state variables at each step, there is more potential to break the problem up into independent subtasks...".

Secondly, there is not a very clear connection between the theory and experiments. For example, Figure 3 has reward rate on the x axis and this seems to have the biggest effect on mixing time (relative to $k$ at least), yet reward rate hardly factors into the theory and is not present in the bounds. The reward rate also seems to affect the results and conclusions as there is not a consistent trend in Figure 3 or 4 across al reward rates. Figure 5 is a highlight of the experiments for me as the distinction between random and learned policies has a clearer connection to strongly connected components and the types of variables present in the theory. So once again the paper does have moments where things are clear and the connections more evident. There just appears to be a need for more consistency. Even on the foundation model results - as necessary as they are - it appears in this case the main point is to just say that foundation models need long enough contexts to disambiguate the various policies they were trained on and that this means they will have a long mixing time. If this is indeed the point, then I think this needs to be clearer as the main point and connection to the theory. Finally, no empirical evidence for the tightness of the bounds is given and typically this is useful and necessary as a source of intuition on how closely the theory fits with what can be expected in practice.

**Questions:**

I have asked some questions in the context of the review above. I have no further questions for the moment, but would appreciate if those questions could be answered.

The main points I need clarified are those raise under the Quality heading in Weaknesses. If these are addressed then I would increase my score to advocate acceptance.

**Limitations:**

I would highlight the need for proof sketches again here, as well as empirical evaluations of the bounds. Both of which are necessary to establish the limitations and intuition of the theoretical results while reading.

---

> ### Author Rebuttal · Authors · 2024-08-07
>
> Thank you for your detailed review of our paper. We very much appreciate your kind words regarding the originality, quality, clarity, and significance of our work. We have attempted to address your key concerns and all points of confusion mentioned in your review below.
>
> **Definitions of aperiodic and unichain:** Thank you for pointing out that this can be a potential source of confusion for readers. We actually did have a relevant part of the appendix where we go through the assumptions and implications of these assumptions for the theoretical findings of our paper at the beginning of Appendix A. For example, we did discuss the implications of the unichain assumption in this section (line 478). However, we can also definitely take this opportunity to provide detailed definitions for all terminology used that we did not have space for in the main text. We will also make sure to direct readers to this section.
>
> **Proof Sketches:** Thank you for your feedback on this point. We didn’t include proof sketches in the submitted draft due to space constraints, but we agree that this can greatly improve comprehension of our results for readers without expecting them to go through our detailed proofs in the appendix. You are right that the reason we mentioned the coupling time was because of its importance in the proof of Theorem 1, but we did not make this link clear enough in the main text. Likewise, the causal parent state variables play a key role in the proof of Theorem 2 and this link was not clear enough in the main text. We believe that providing these sketches is a great idea and will let us better explain to readers why we introduce some important notation, terminology, and assumptions.
>
> **Connections to $\beta$ and $g$ in Figures 1 and 6:** We really like the idea to provide explicit information on $\beta$ and $g$ in the description of these figures. See below when we discuss bound tightness an example of these values for Figure 1. In section B.1 of the appendix of the submission, we did make an initial attempt to explain state variable structure of these domains, but it would definitely also be help to take it a step further to make the connection with our theoretical findings more concrete.
>
> **$\ell$ in Theorem 1:** Thank you for bringing this up. We introduced $\ell$ chiefly to contrast with the prior result from Kearns and Kholler (line 153). It would indeed be better to introduce $\ell$ at this point after the theorem statement.
>
> **$k$ in Theorem 2:** Thank you also from bringing this up. On a similar point reviewer kzFL also suggested that it could be useful to provide line 663 from the appendix to make the connection with $k$ clearer in the main text. We will make this clear in the statement of Theorem 2 and subsequent proof sketch in the final draft.
>
> **Reward rate hardly factors into the theory:** In the theory we mostly do not mention reward rates in order to keep our results applicable to general $\epsilon$-mixing times. However, our experiments are highly dependent on the reward rate because we use the $\epsilon$-return mixing time in order to not overly focus on spurious state features. $\epsilon$-return mixing time itself is fundamentally grounded in the reward rate (line 108).
>
> **Consistency of trends in Figures 3 and 4:** There is a difference between the input required to express a policy and the input available to a policy. For example, let’s consider the irrelevant variable example from Figure 1 where it is only necessary for the policy to leverage $x$ and not $y$. Because the optimal policy only depends on $x$, this policy is included within the set of policies that only depend on $x$. However, the optimal policy is also included within the set of policies that take both $x$ and $y$ as input. Even though the functional form of this policy depends on both $x$ and $y$, the actions of this particular policy are only conditionally dependent on $x$. So regardless of the functional form used to search for the optimal policy, the optimal policy is the same in both cases. It receives the same reward rate and encounters the same mixing time. As a result, the optimal reward rate in Figures 3 and 4 is associated with the same optimal policy regardless of the context length and model class. What is more interesting is the mixing times associated with policies experienced along the way for each model class.
>
> **Evidence of bound tightness:** Thank you for this great suggestion. We will add an analysis of bound tightness to section 2.3 for the problems highlighted in Figure 1. In every case $\beta=0.1$. In the irrelevant variables example, $g=1$ for policies that only condition on $x$, and $g=2$ for policies that condition on both $x$ and $y$. As a result, the bound should be 10 times tighter for the set of policies that only condition on $x$ than the set of policies that condition on both $x$ and $y$. In practice the maximum $\epsilon$-return mixing time is 7.6 times smaller. In the independent subtasks example, $g=2$ for policies that always condition on $x$, $y$, and $z$. Meanwhile, $g=1$ for policies that focus on the relevant variable $x$ or $y$ depending on the value of $z$. So the bound should once again be 10 times tighter, but the maximum $\epsilon$-return mixing time is 3.6 times smaller in this case. One reason that we believe these bounds are not as tight as possible for this experiment has to do with using the $\epsilon$-return mixing time that is not sensitive to state changes that don’t have an impact on the reward. In these examples, we adopted a simple sparse reward where many states get a reward of 0. For the final draft we will also include a variant of the experiment where the reward is set to be different at each state such that the Markov chain over states must fully mix for the $\epsilon$-return mixing time to also mix. We expect that this setting will verify these bounds to be even significantly tighter than what we have measured here in practice.

---

> > ### Comment · Reviewer_hXkG · 2024-08-11
> > **Response to Rebuttal by Reviewer hXkG**
> >
> > I thank the authors for their thorough response. My following thoughts are as follows:\
> > **Definitions of aperiodic and unichain**: Yes, explicitly pointing to the appendix here is necessary and will address my concern.\
> > **Proof Sketches**: Would the authors be able to provide these before the end of the discussion period? I appreciate that I have responded a couple of days into the discussion and so if these are still draft version I will be happy. It just seems like accepting this change without review is unwise.\
> > **Connections to $\beta$ and $g$ in Figures 1 and 6, $l$  in Theorem 1, $k$  in Theorem 2**: Noted with thanks.\
> > **Reward rate hardly factors into the theory**: I understand better now. Thank you. I do think this is a general weakness of the work as it stands as clearly in the experiments reward rate is having a material effect. Considering and experimental design or metric which does not have this issue or incorporating reward rate into the theory would help. However, I do not think this is grounds for rejection, subject to a clearer mention of the link in a revised draft. I would just put this under the "why not higher" part of my final score rather than a "reject vs accept" point.\
> > **Consistency of trends in Figures 3 and 4**: I appreciate the point the author is making. Correct me if I'm wrong then, but I think the point is that the nuisance input variable can change the mixing time but not optimal policy. If so, I understand this, but I'm still not clear how this addresses the fact that the reward rate can change the relative mixing time between context lengths (Fig 3) or architectures (Fig 4). For Fig 3 the switch between $k=4$ and $k=2$ is right at the end and within the variance bounds, so I could drop this point there. But I'm still unsure why reward rate can flip the findings.\
> > **Evidence of bound tightness**: This is an important addition. As can be see from those results, qualitatively the results hold but quantitatively there can be a big difference. The $3.6$ shrink rather than the expected $10$ is a bit of a concession. But I think the qualitative results are still meaningful and significant, so my main issue here is that it need to be added. I trust it will be in the subsequent draft. I notice that the authors did not use the figures page. If they are still able to and can provide results for the proposed experiment: "include a variant of the experiment where the reward is set to be different at each state" that would be helpful. However, without reviewing these results it is difficult to let it influence my assessment.
> >
> > For the moment, I feel I am understanding the work better and will raise my confidence score. My rating will remain unchanged but if the authors provide the draft **proof sketches** and a bit more clarity on **Consistency of trends in Figures 3 and 4** I *will* increase my score to advocate acceptance.

---

> > > ### Author Response · Authors · 2024-08-14
> > > **Re: Response to Rebuttal by Reviewer hXkG (Part 1)**
> > >
> > > Thank you for providing such a thorough response to our rebuttal. We really appreciate your willingness to work with us constructively to improve our paper during the discussion period. As you suggested, we have provided first draft proof sketches for Theorems 1 and 2 using the notation we have established in the main text.
> > >
> > > **Proof Sketch for Theorem 1:** Due to the sorting of the $\ell$ strongly connected components, our analysis is based on coupling each of the $\Gamma^\pi_i$'s in succession. Because it is possible that multiple $\Gamma^\pi_{i}$’s couple at the same step, every step where the Markov chain does not fully couple must be a step where some $\Gamma^\pi_{i}$ does not couple. Our proof proceeds in the following high-level steps:
> > >
> > > 1.	The probability of $\Gamma^\pi_{i}$ coupling at a given step once $\Gamma^\pi_1, ..., \Gamma^\pi_{i-1}$ have all already coupled is $\geq \beta^g$.
> > >
> > > 2. Thus the probability of $\Gamma^\pi_{i}$ not coupling at a step when $\Gamma^\pi_1, ..., \Gamma^\pi_{i-1}$ have all already coupled is $\leq (1 - \beta^g)$.
> > >
> > > 3.	So the joint probability of $\Gamma^\pi_{i}$ not coupling for $m_i \geq 0$ steps when $\Gamma^\pi_1, ..., \Gamma^\pi_{i-1}$ have all already coupled is $\leq (1-\beta^g)^{m_i}$.
> > >
> > > 4.	If $\tau > m$, then $\sum_{i=1}^\ell m_i = m$ and the joint probability that $m$-steps have been spent not coupling in some $\Gamma^\pi_{i}$ has a probability bound independent of the particular allocation of $m$ into individual $m_i$. Thus we can conclude that $P(\tau > m) \leq (1-\beta^g)^m$.
> > >
> > > 5.	Leveraging the identity that $1-x \leq e^{-x}$ for $x \geq 0$, we find that $P(\tau > m) \leq (e^{-\beta^g})^m$.
> > >
> > > 6.	The Markov chain is $\epsilon$-mixed if $P(\tau > m) \leq \epsilon$, so it must be $\epsilon$-mixed if $(e^{-\beta^g})^m \leq \epsilon$, which implies that it is  is $\epsilon$-mixed if $m \geq \frac{1}{\beta^{g}} log(1/\epsilon)$.
> > >
> > > 7.	Finally, we note the relationship between $t_{ret}^\pi(\epsilon)$ and $t^\pi_{mix}(\epsilon)$ following Lemma 1 of [1].
> > >
> > > **Proof Sketch for Theorem 2:** Lemma 1 in the appendix considers the mixing time relationship of policy classes conditioned on subsets of the state variables that other policy classes are conditioned on. Our proof proceeds by applying our notation from Section 3 in which $k' \geq k$ for all $t$ to the results of Theorem 1 and Lemma 1:
> > >
> > > 1. We consider the causal parent state variables of each observation, action, and reward to conclude that
> > > $Par(h^{(k)}) \subseteq Par(h^{(k')})$, which implies that $n \geq n_{k'}(t) \geq n_k(t)$.
> > >
> > > 2.	Through Lemma 1 we show that by rule of Cartesian products over subsets $0 \leq \beta_{k'} \leq \beta_k \leq 1$.
> > >
> > > 3.	Through Lemma 1 we also demonstrate that $g_{k'} \geq g_{k} \geq 1$ because causal connections in $\mathcal{D}^\pi$ are only added and not removed when the context length is increased.
> > >
> > > 4.	This then implies that $1/\beta_{k'}^{g_{k'}} \geq 1/\beta_{k}^{g_{k}}$, which is sufficient to prove Theorem 2 using Theorem 1 because $\epsilon$ is independent of $k$.
> > >
> > > Definitely let us know whether these proof sketches helped provide clarity for you. We are happy to revise them based on your feedback as this is just an initial draft.
> > >
> > > [1] Michael Kearns and Satinder Singh. Near-optimal reinforcement learning in polynomial time. *Machine learning,* 49(2):209–232, 2002.

---

> > > > ### Author Response · Authors · 2024-08-14
> > > > **Re: Response to Rebuttal by Reviewer hXkG (Part 2)**
> > > >
> > > > **Consistency of trends in Figures 3 and 4:** Thank you for pointing out that this is still confusing. Our overall point is that at some reward rates (i.e. the optimal one) there is only one potential policy that achieves that reward rate that can be learned. When the policy of two different architectures or context lengths learn the exact same behavior, both their reward rates and mixing times should be expected to be the same. What is more interesting is analyzing the difference between the mixing times when the policies for two different architectures or context lengths arrive at a similar reward rate while still displaying different behavior (i.e. the policies learned are not exactly the same). We would argue that we don’t actually see instances of findings flipping in our experiments. Rather, we see two trend that may appear as inconsistency, but really are perfectly consistent with the theory of our paper:
> > > >
> > > > 1.	At some reward rates multiple context lengths or architectures learn the exact same policy and estimated mixing times are the same or their differences are not statistically significant.
> > > > 2.	Sometimes a context length or architecture will learn to achieve a value of the reward rate that other policies and architectures never learn to achieve. In this case there is not a clear basis for comparison at these values.
> > > >
> > > > Please also let us know whether this explanation helps clarify your confusion on the trends in Figures 3 and 4. While at some points in the figures multiple architectures or context lengths arrive at the same solution or are not directly comparable, when they are comparable and different, we see consistent orderings between the different policy classes considered.
> > > >
> > > > **Additional Experiments on Bound Tightness:** Because of your interest in this experiment, we thought you may like to hear about our intermediate progress. We have tried a setting of the irrelevant variables example where reward is still $1.0$ when $x = x0$, but edited to $-0.25$ when $x = x1$ and $-1.5$ when $x = x2$ rather than $0$. This setting was chosen to have no impact on the optimal policy while densifying rewards without making a substantial change to the magnitude of best reward rates achieved. The ratio between the maximum mixing times is now improved to 8.8 times smaller when only focusing on the relevant variable, rather 7.6 times smaller with the previous sparse rewards. Note that our reward still has no dependence on the irrelevant variable $y$ and is thus invariant to a large part of the change in the state space. We will also try changing the reward based on $y$ in a way that does not impact preferences over policies in order to close the gap further between $t_{ret}^\pi(\epsilon)$ and the mixing time over the full state space $t^\pi_{mix}(\epsilon)$. We believe this is likely to push our results even closer to the theoretical bound of a 10 times ratio for this problem. We definitely appreciate why you are more interested in the independent subtasks example, but we were unfortunately not able to get new results in time for the close of the discussion phase. As discussed in Appendix B.1, the search over the policy space is very expensive for that problem, so it was not feasible to complete in time. We certainly understand if you feel that these results are too incomplete to impact your review, but still wanted to speak to our progress just to give you a better overall picture of where things stand.

---

### Official Review · Reviewer_kzFL · 2024-07-13

**Soundness:** 3
**Presentation:** 2
**Contribution:** 3
**Rating:** 5
**Confidence:** 3

**Summary:**

This work analyzes the mixing time of a policy in average-reward reinforcement learning problems. It shows that the mixing time is related to the structure of the underlying dynamic Bayesian network (DBN), specifically how the state variables are grouped into strongly connected components by the policy. Additionally, the context (trajectory history) that is fed to a policy has an impact on the mixing time: the longer the context length, the harder to mix. The mixing time is empirically verified in several toy problems. The paper also demonstrates that the Transformer architecture is more prone to using longer context and thus harder to mix.

**Strengths:**

- Theoretically analyze the mixing time for the average-reward setting and its relation to the input context length
- Clear writing and presentation in most places
- Empirical verification using modern model architectures

**Weaknesses:**

Multiple places require further clarification.

1. It is challenging to connect Fig.1 to the main point of L167-174 (as well as Thm.1). For example, it is unclear what $\beta,g$ are in this example and how the bound in Thm.1 is related to the estimated mixing time. This remains unclear even after reading the details in the appendix.

2. L243: It is unclear what’s the difference between $\Theta$ and $\Theta’$.

3. To better interpret Thm.2, it would be helpful to point out how $\beta_k$ and $g_k$ evolves as $k$ increases in the main text. This is discussed in the appendix (L663) but it is important and it would be better to see it in the main text instead.

4. L286: "It appears that attention mechanisms make it easier to focus on the full context rather than i.e. only the recent parts and that this capability is predictably enhanced when the model capacity is increased." It will be helpful to visualize this, e.g., seeing how dense the attention map is w.r.t. different reward rates.

**Questions:**

Please refer to the weakness above.

**Limitations:**

The authors adequately addressed the limitations

---

> ### Author Rebuttal · Authors · 2024-08-06
>
> Thank you for your thoughtful review of our paper. We appreciate that you highlighted our theoretical contribution and want to thank you for your praise regarding the writing quality and empirical verification of our theoretical results. Below we have tried to provide clarity regarding each of your questions in the weaknesses section. We hope that you will consider raising your score if we have adequately addressed your concerns.
>
> **Question 1:** Thank you for pointing out your confusion regarding $\beta$ and $g$ in Figure 1. In the lines between 180 and 213 we went over the main facts needed to derive these terms in combination with the figure caption, but you are definitely right that it is much better to state them clearly as readers will not be as familiar with this kind of analysis and may find it difficult to follow along. In every case $\beta=0.1$. In the irrelevant variables example, $g=1$ for policies that only condition on $x$, and $g=2$ for policies that condition on both $x$ and $y$. In the independent subtasks example, $g=2$ for policies that always condition on $x$, $y$, and $z$. Meanwhile, $g=1$ for policies that focus on the relevant variable $x$ or $y$ depending on the value of $z$. We will be sure to substantially update this section in the final draft to eliminate any potential confusion.
>
> **Question 2:** Thank you for mentioning that line 243 was confusing. Our motivation for introducing the prime notation was simply to draw a distinction between the parametric class of policies over the state space $\theta’ \in \Theta’$ and policies over interaction history windows $\theta \in \Theta$. The point being that the set of possible policies are not the same or directly comparable due to the different functional form resulting from different input spaces. We will add a footnote to the final draft to clarify this point.
>
> **Question 3:** Thank you for this insightful comment highlighting this potential area of confusion. We will definitely include this context about how $\beta_k$ and $g_k$ evolve with $k$ from line 663 of the appendix in the main text to better explain Theorem 2. We should not have assumed readers will figure this out themselves as this provides unneeded cognitive load when interpreting Theorem 2 and we want the results to be easily accessible to as wide as audience as possible.
>
> **Question 4:** Thank you also for you comment about visualizing attention maps to validate our conclusion on line 286. This is a really fabulous suggestion in order to validate whether this intuition we presented actually is consistent with the results. Unfortunately, we did not log data related to attention when we conducted these experiments and doing so will require us to rerun the experiments highlighted in Figure 8 from scratch. This corresponds to 1,000 individual experiments with specific seeds, requiring significant computational resources and time. For the final draft we will rerun these experiments while keeping track of the average attention attributed to each element of the interaction history window across the 8 heads in each layer. It will be interesting to see if the spread of attention is indeed more even on average for Transformer models that experience higher average mixing times. Theorem 2 shows theoretically that what leads to potentially high mixing times is when our model is highly sensitive to a large part of the interaction history at all times. Following from this insight, it seems natural that we may see higher mixing times for Transformers than LSTMs and for bigger Transformers than smaller Transformers. However, we definitely agree that this conclusion will be much stronger if each element of this hypothesis is empirically validated. We really appreciate this suggestion and are excited about being able to provide more clarity about the mechanisms at play in Figure 8 for the final draft.
>
> Please let us know if we can provide any further clarity regarding any of these points during the discussion period.

---

> > ### Comment · Reviewer_kzFL · 2024-08-12
> >
> > I thank the authors for the clarifications. The paper can help reveal some potential issues theoretically, but the presentation and experiments can be better so I would like to maintain my score.

---

> ### Author Response · Authors · 2024-08-12
>
> Thank you for participating in the discussion period and for your constructive feedback that we believe has helped improve our paper. We were wondering if you could clarify further about what concerns regarding the presentation and experiments of our paper still remain after our response to your review. We just ask because many of your comments, especially regarding the presentation, seemed straightforward for us to address. We want to make sure that we understand how our paper can be improved to address your recent feedback.

---

### Official Review · Reviewer_LbpG · 2024-07-13

**Soundness:** 4
**Presentation:** 4
**Contribution:** 4
**Rating:** 7
**Confidence:** 2

**Summary:**

In the quest for applying RL to more realistic tasks in the long term, the authors interrogate the relationship between context length and mixing times. They provide an extensive theoretical analysis leading to a novel finding regarding the trade-off between learning with large context and slower evaluation. The authors also perform empirical evaluation on a toy-example as well as Minigrid environments to provide evidence of the proposed theories.

**Strengths:**

The work is novel, providing both a theoretical and empirical analysis. The toy example provides evidence for high mixing times and is followed by analysis in larger scale environments: the experimentation is comprehensive. This is a significant contribution and the paper is well-written.

**Weaknesses:**

While the theoretical contribution is the main contribution, the main paper should ideally include more empirical evidence which has been moved to the supplementary material. It would be ideal if this was extensively explored in a larger journal paper.

**Questions:**

I am interested in hearing the author's thoughts on what architectural changes could be introduced to tackle the context length and mixing times trade-off

**Limitations:**

The work discusses the limitations of existing approaches. While the work is essential, it could only be possible with significant compute resources.

---

> ### Author Rebuttal · Authors · 2024-08-05
>
> We wanted to begin by thanking you for the kind words in your review regarding a number of aspects of our paper including the novelty, comprehensive empirical analysis, and writing quality. We really appreciate this validation of our work. We also wanted to make sure to engage with all of your concerns and suggestions.
>
> **Experiments in Appendix:** We definitely agree that space limits us in being able to provide our experiments in the main text, but we do feel that it should be possible to provide a concise overview of the key findings in the space limitations of a standard conference paper. This would allow us to reach the biggest audience possible (given that a much smaller community would be interested to read through an extended version). We were wondering if there were any experiments in particular that you felt would make a significant addition to the main text. If so, we would be happy to see if the paper could be restructured to accommodate for this.
>
> **Possible Architectural Changes:** We really appreciate this question as it cuts to the chase of the core motivations of our work. Most work on RL that even acknowledges the challenges of high mixing times does so with a defeatist mentality, assuming that problems with high mixing times are unavoidably harder and that there is basically nothing that could be done about it. If our work makes any contribution to the community, we hope it will be to highlight that this isn’t actually true and that the policy class we choose to optimize over itself can have a big impact on mixing properties. We hope that readers of our paper come away from it asking exactly the question that you did.  What our paper shows in Theorem 2 is that what leads to potentially high mixing times is when our model leverages a monolithic representation that is highly sensitive to a large part of the interaction history at all times. This is particularly descriptive of how generic transformers work, but there are multiple already existing research directions that seem well suited to scaling to high context lengths while providing less history sensitivity at each step:
>
> - **Hierarchical RL:** In hierarchical RL frameworks such as options [1] it should be possible in domains with temporal coherence to approximate a policy with a longer context length by multiple sub-policies with a smaller context length. This is particularly relevant for approaches to option learning that learn an independent lower dimensional representation space for each option as in [2].
>
> - **Hybrid Transformer Working Memory Architectures:** Recent work has aimed to improve the effective context length of transformers by augmenting them with some kind of bounded working memory component [3, 4, 5]. While these papers are typically solely motivated by computational efficiency, they effectively serve the role of extending Transformers to longer contexts while making them less sensitive to experiences in this context that are not reflected in the memory. This effectively bring Transformers closer to some of the incremental design patterns of RNNs, which our experiments indicate experience lower mixing times. This makes sense because they must commit to a single representation of the history at each step and do not have the capability to constantly reassess their history representation dynamically as the context changes the way Transformers do. As such, even state-space models [6, 7, 8] or hybrid alternatives could be attractive in achieving the performance of Transformers while potentially alleviating mixing time concerns during learning.
>
>  - **Tracking Policies:** If we aim to learn a non-stationary tracking policy solution concept as argued by [9] we could potentially represent a stationary policy over a longer context length with a non-stationary set of small context length policies. In effect, this becomes similar to what is achieved by the hierarchical RL policies described above. One additional subtlety to highlight in this case is that this solution is also limited by the mixing properties of the tracking policy parameters, so it would also be necessary to tune this approach to move through parameter space as fast as possible.
>
> We will make sure to add this additional detail into the discussion section of the final draft where possible.
>
> [1] Sutton, Richard S., Doina Precup, and Satinder Singh. "Between MDPs and semi-MDPs: A framework for temporal abstraction in reinforcement learning." Artificial intelligence 112.1-2 (1999): 181-211.
>
> [2] Abdulhai, Marwa, et al. "Context-specific representation abstraction for deep option learning." Proceedings of the AAAI Conference on Artificial Intelligence. Vol. 36. No. 6. 2022.
>
> [3] Pham, Kha, et al. "Generative pseudo-inverse memory." International Conference on Learning Representations. 2022.
>
> [4] Das, Payel, et al. "Larimar: Large Language Models with Episodic Memory Control." Forty-first International Conference on Machine Learning. 2024.
>
> [5] Munkhdalai, Tsendsuren, Manaal Faruqui, and Siddharth Gopal. "Leave no context behind: Efficient infinite context transformers with infini-attention." arXiv preprint arXiv:2404.07143 (2024).
>
> [6] Gu, Albert, and Tri Dao. "Mamba: Linear-time sequence modeling with selective state spaces." arXiv preprint arXiv:2312.00752 (2023).
>
> [7] Dao, Tri, and Albert Gu. "Transformers are SSMs: Generalized models and efficient algorithms through structured state space duality." arXiv preprint arXiv:2405.21060 (2024).
>
> [8] Samsami, Mohammad Reza, et al. "Mastering Memory Tasks with World Models." The Twelfth International Conference on Learning Representations. 2024.
>
> [9] Sutton, Richard S., Anna Koop, and David Silver. "On the role of tracking in stationary environments." Proceedings of the 24th international conference on Machine learning. 2007.

---

> ### Comment · Reviewer_LbpG · 2024-08-11
> **Response noted**
>
> Thank you for engaging with the review. I believe adding the discussion on architectural changes to the paper is essential. I think that space could be better used through subfigures so that Figure 5 and Figure 8 could be placed side by side, for example. I will stick to my original score.

---

### Official Review · Reviewer_bcCn · 2024-07-14

**Soundness:** 2
**Presentation:** 2
**Contribution:** 1
**Rating:** 4
**Confidence:** 4

**Summary:**

Authors analyzed the relationship between mixing time (policy evaluation time) with increase in context length for non-markovian, POMDPs. Certainly, increasing context, increases mixing time.

**Strengths:**

Authors propose a tighter upper-bound for mixing times of multi-dimensional mdp with longer contexts

**Weaknesses:**

Reviewer is not clear why it is of any significance, this seems like a trivial observation that may not even need a theoretical justification.
Reviewer is not sure whether it adds any new knowledge to the field.

As the context length is increased, we increase dimensionality of the problem, certainly increases the cardinality, so the reviewer doesn't find this result of any significance, unless reviewer is in-correct with his understanding of the draft.
I don't intend to dis-respect the amount of work put in by the authors, however, I merely wonder how this direct observation needs a (tighter) upper bound, ideally in the current scenario, lower bound is of more importance. Experimental evaluation also doesn't offer any key insights, since context lengths considered are quite small (up to 100)

I have looked at main draft thoroughly and did not look at the appendix.

**Questions:**

See the above

**Limitations:**

See the above

---

> ### Author Rebuttal · Authors · 2024-08-07
>
> Thank you for taking the time to review our paper. We are sorry to see that you are so negative about the value of our contributions to the community. We believe that there are some key points of confusion that may have led you to this perspective that we will attempt to address below. We will also reference other reviews when helpful to contrast with other perspectives on our paper.
>
> **Our contribution:** After reading through your summary of our contributions, we would like to clarify a potential confusion about the novelty of our results. Our result in Theorem 1 does indeed provide a tighter bound on a prior result, but this is only an intermediate result of our paper necessary for proving Theorem 2, which is our main contribution. Theorem 2 is entirely novel. We are not aware of any paper from the prior literature that has even mentioned the fact that the context length of the interaction history sent to a policy would have an impact on its mixing time. When the reviewer writes “I merely wonder how this direct observation needs a (tighter) upper bound,” the answer is that the tighter bound of Theorem 1 is essential for being able to prove the novel result of Theorem 2.
>
> **Trivial observation:** This comment seems at the heart of your criticisms of our paper. As such, we will attempt to break down your argument piece by piece to provide you with clarification on why this may be much harder to show than you realize. The first aspect of your argument “As the context length is increased, we increase dimensionality of the problem” requires some immediate clarification. Often in RL when we talk about “the dimensionality of the problem,” we are talking about the size of the environment. However, increasing the context length sent to a policy has no direct impact on the environment. Indeed, prior work at this conference [1] has established that as environments are scaled up, their mixing time also increases. We must reiterate though that the environment is not scaled up when the context length is increased, so this result is not at all related to our theoretical contribution. The reviewer also mentions that increasing the context length “increases the cardinality, so the reviewer doesn't find this result of any significance.” The cardinality that is being increased here is the set size of the possible interaction histories $h^{(k)}_t$. But we do not see how the reviewer jumps from the fact that this set size is increasing to the fact that the mixing time is going up. Notice that this $\epsilon$-mixing time is defined in terms of states and not observations, so it is not obvious that an input that does not explicitly include the state being sent to a policy will impact the $\epsilon$-mixing time. We know that the reviewer mentioned not looking through the appendix, but the proofs of Theorem 1 and 2 are not direct results of cardinality arguments, rather they have to do with the underlying causal structure of the Markov chain induced by the policy in the environment.  If the reviewer has a better idea of how to show this result in a trivial way that goes beyond a vague intuition, we would kindly request that they spell it out in more detail. As suggested by reviewer hXkG, we will provide proof sketches to aid comprehension in the main text of the final draft.
>
> **Bound Tightness:** As other reviewers also suggested, in the final draft we will add an analysis of bound tightness to section 2.3 for the problems highlighted in Figure 1. In every case $\beta=0.1$. In the irrelevant variables example, $g=1$ for policies that only condition on $x$, and $g=2$ for policies that condition on both $x$ and $y$. As a result, the bound should be 10 times tighter for the set of policies that only condition on $x$ than the set of policies that condition on both $x$ and $y$. In practice the maximum $\epsilon$-return mixing time is 7.6 times smaller. In the independent subtasks example, $g=2$ for policies that always condition on $x$, $y$, and $z$. Meanwhile, $g=1$ for policies that focus on the relevant variable $x$ or $y$ depending on the value of $z$. So the bound should once again be 10 times tighter, but the maximum $\epsilon$-return mixing time is 3.6 times smaller in this case. One reason that we believe these bounds are not as tight as possible for this experiment has to do with using the $\epsilon$-return mixing time that is not sensitive to state changes that don’t have an impact on the reward. In these examples, we adopted a simple sparse reward where many states get a reward of 0. For the final draft we will also include a variant of the experiment where the reward is set to be different at each state such that the Markov chain over states must fully mix for the $\epsilon$-return mixing time to also mix. We expect that this setting will verify these bounds to be even significantly tighter than what we have measured here in practice.
>
> **Adding Knowledge to the Field:** We should note that reviewer FWjm also felt that the result in Theorem 2 was intuitive, but thought it was still a contribution to provide the formal analysis and that it was interesting to link context lengths with mixing times. Reviewer LbpG also felt that the contribution was significant. Likewise, reviewer hXkG felt that the contribution had a high degree of significance and that the paper’s biggest strength was its potential influence on subsequent work because its insights are timely. Moreover, reviewer aEyh praised our new angle at an important and interesting problem, remarking that the “community will benefit from this work being available to them.”
>
> **Insights from Evaluation:** Again we should note that reviewer FWjm called our evaluation convincing and reviewer LbpG felt the experiments were comprehensive.
>
> [1] Riemer, M., Raparthy, S.C., Cases, I., Subbaraj, G., Puelma Touzel, M. and Rish, I. “Continual learning in environments with polynomial mixing times.” Advances in Neural Information Processing Systems, 2022.

---

### Official Review · Reviewer_FWjm · 2024-07-18

**Soundness:** 3
**Presentation:** 3
**Contribution:** 3
**Rating:** 6
**Confidence:** 2

**Summary:**

I am not a theorist and so did not feel able to provide a detailed review of the theoretical parts of this work.

The authors discuss the trade-off between context length and mixing time of MDPs, demonstrating that a longer context length leads to longer mixing times and hence greater difficulty in evaluating a given policy. This is an intuitive observation -- if my next action conditions on an observation many timesteps ago, I must wait at least that many timesteps before such a dependency can even be shown, let alone reliably evaluated. However, the authors introduce novel theoretical analysis formally demonstrating this connection. They then perform empirical experiments providing evidence validating their theoretical findings in simple settings and early evidence that transformers have a higher mixing time than other architectures.

**Strengths:**

* The paper is very well written. Despite my lack of theoretical RL knowledge, I was able to follow the arguments of the paper well.
* The implications of their argument that there is a link between context length and mixing time are potentially interesting.
* The evaluation of their theoretical findings is convincing, with Figure 3 nicely demonstrating their central result.

**Weaknesses:**

* It is not clear to me that the larger mixing times are a significant problem in practice. The authors provide a rough intuition of when high mixing times would be relevant, but this example is not grounded in a real-world benchmark or evaluation. This makes it impossible to assess the practical impact of the authors' theoretical observations for policy evaluation, especially given the toy nature of the environments they evaluate on.
* I'm not sure i understand the point being made by Figure 9. The authors show different values of context length, $k$, and show that for $k=25$ good performance can be attained, but for lower or higher values of $k$ performance degrades. Is this just a straightforward example of over/underfitting? I'm not sure what this has to do with mixing times? Clearly for most problems $k$ will have an optimal value much higher than the $k$ value of the ensemble of poicies used to generate the data, but I'm not sure I understand what this demonstrates about evaluation.

**Questions:**

See weaknesses

**Limitations:**

The authors discuss the limitations of their theoretical analysis in the paper. It would be nice to flesh out this discussion with more examples of concrete environments or settings where their theoretical analysis is relevant or not.

---

> ### Author Rebuttal · Authors · 2024-08-05
>
> We wanted to begin by thanking you for your thoughtful review of our paper. We really appreciate your praise of the implications of our theoretical findings, the evaluation of these findings, and the writing quality of our paper. We also found your intuitive explanation of our findings to be quite insightful. We really like the way you put it and will be sure to reference this kind of explanation in the final draft.
>
> **Confusion Regarding Figure 9:** Thank you for pointing out to us how confusing this was in our submission. What wasn’t clearly articulated in our description from lines 316 to 317 was that this experiment was really motivated as a response to a particular potential criticism of our analysis in section 4. One could potentially argue that while a larger context length is needed to fit the set of behavior policies than each had individually, this is simply “overfitting” to the training data and not useful in achieving downstream performance. Our intention in Figure 9 is to provide a concrete counter example to refute this argument. What we believe must have made this even more confusing is that our main point in Figure 9 could only be understood by reading the in-text descriptions of the training accuracy at each context length value. In the final draft, we will update the label to include the training accuracy corresponding to each value of $k$. Figure 9 then chiefly shows that the models with the best generalization have 100% training accuracy. You are also right that another trend highlighted in Figure 9 is that $k$ values that are unnecessarily large seem more prone to overfitting, which isn’t very surprising. We believe these points are interesting to highlight, but also acknowledge that they are only indirectly connected to mixing times through the findings of our paper and simply serve as supporting evidence for the need for growing the context length when building foundation models.
>
> **Practical Impact:** Thank you for mentioning your confusion about the relevance of high mixing times and our analysis in practice. We believe the settings of greatest relevance to our work are those related to continual or multi-task environments where agents are evaluated as generalists over a number of skills rather than just solving a single narrow task. As such, we believe that focus on the difficulties presented by high mixing times is timely in the age of foundation models. As mentioned at the end of section 3.1, our analysis will not have relevance in problems where there are few state variables that each impact every observation. However, composite tasks that test a number of sub-skills naturally tend to have many total state variables with relatively few impacting each observation. So, for example, simple Atari domains such as Pong and Breakout will not suffer from high mixing times, but i.e. continual learning over multiple Atari games including Pong and Breakout will as a result of the sparsity of causal impact of variables across games. Another example would be open world environments that feature a number of natural sub-tasks and sub-regions such as RPGs like the Pokémon series of games or the Legend of Zelda games. Moreover, broadly speaking AI assistant tasks that include providing help on a number of topics rather than just one should also suffer from issues with high mixing times. We will provide more detail along these lines in the final draft. In the submitted draft, we wanted to prioritize focusing on our novel results, but it probably would help readers to contextualize our analysis more deeply with recent prior work on this topic such as [1]. In this paper it was shown why continual learning tasks with multiple subtasks have high mixing times and in-depth experiments on mixing times were conducted for the Atari games. It was found that the common setting in the literature results in practical $\epsilon$-return mixing times on the order of 10 billion time steps even when only considering 6 Atari games. [1] also argues theoretically how high mixing times are a primary driver of poor performance and training instability on continual learning tasks. In light of these prior results in the literature and the complexity of analyzing large scale and long horizon games, we opted for smaller scale experiments in our paper to allow for greater control and more interpretable insights.
>
> [1] Riemer, M., Raparthy, S.C., Cases, I., Subbaraj, G., Puelma Touzel, M. and Rish, I. “Continual learning in environments with polynomial mixing times.” Advances in Neural Information Processing Systems, 2022.

---

> > ### Comment · Reviewer_FWjm · 2024-08-12
> > **Response**
> >
> > Thanks very much to the authors for their rebuttal and clarifying comments. I will maintain my score for now.

---

### Author Rebuttal · Authors · 2024-08-07

We wanted to begin by thanking all six reviewers assigned to our paper. We were very happy to see so many positive comments about our work. We were also grateful to see so many constructive comments that will help make our work even better in the final draft. We have provided a detailed rebuttal for each reviewer in order to address all questions and concerns mentioned in their reviews. We hope that if there are still any concerns that reviewers feel were not sufficiently addressed, we will have an opportunity to follow up on these points during the discussion period.

A benefit of receiving such a large and diverse set of reviews is that we also received multiple follow up questions unique to each reviewer. These questions can generally be addressed with relatively modest adjustments to the presentation of the paper from the submitted draft. However, there were also a few topics that multiple reviewers asked about:

- **Dependence on reward rate in Figures 3 and 4:** In the responses to each reviewer, we tried to provide some context to help them understand why it is expected that the $\epsilon$-return mixing time will depend on the reward rate and why it is expected for mixing times to converge across different models at particular values of the reward rate. We believe that some small adjustments to the presentation of the theoretical results in Sections 2 and 3 will be sufficient to make these points clearer for future readers.

- **Evidence of bound tightness:** By providing additional background required for analyzing bound tightness for the settings in Figure 1 of Section 2.3, we can provide direct validation of the degree of tightness. Additional experiments would be needed to show if the bound could be empirically verified to be even tighter, but these experiments would be simple variants of the experiments already provided in the submitted draft. As such, we believe it will be straightforward to provide increased clarity for readers on this topic.

- **Connecting the domains considered to the concepts of $\beta$ and $g$:** The information needed to derive this information was already provided for Figure 1 in Section 2 and we will simply make the direct connection clearer in revisions. We will also make edits to Figure 6 and Section B.1 of the appendix so that this connection is clear in our function approximation experiments as well. We had already explained how these environments could be described in terms of state variables, so providing this analysis is a simple extension of our discussion of these environments in the submitted draft.

- **Clarifying the dependence on $k$ in Theorem 2:** As pointed out by reviewer kzFL, this detail is already included in the appendix and it will be simple to provide improved clarity for readers by presenting this alongside Theorem 2 in the main text.

---

### Comment · Area_Chair_8oKR · 2024-08-10
**A quick question for authors**

Can you please define $\rho(\pi, s, m)$ mathematically in terms of rewards? Thank you.

---

> ### Author Response · Authors · 2024-08-12
> **Re: A quick question for authors**
>
> Thank you for your question. Intuitively, what we meant was $\rho(\pi, s_0, m) := E_\pi[\frac{1}{m} \sum_{t=1}^m r_t | s_0] $. Maybe it would have been clearer if we used the term reward rather than return?
>
> The $\epsilon$-return mixing time was first proposed in [1] with the authors defining an analogous term to $\rho(\pi,s_0,m)$ in their different notation using definition 4 as well as equations 3 and 5 to accomplish this (see https://www.cis.upenn.edu/~mkearns/papers/KearnsSinghE3.pdf). Their definition is more formal than what we provided above. They begin by defining what they call a $T$ path in the MDP i.e. a particular $T$-step path within the MDP. $T$ here is analogous to what we call $m$ in our paper. Then they define the average undiscounted return per step along that $T$ path in equation 3. This allows them to define a term analogous to $\rho(\pi,s_0,m)$ in equation 5, which is the expected average undiscounted return per step where the expectation is taken over the likelihood that a particular $T$ path is executed by the policy $\pi$ from the starting state $s_0$.
>
> We appreciate you raising the potential confusion regarding $\rho(\pi, s_0, m)$ to our attention and will definitely include a definition in the final draft of our paper. We were hoping to ask your opinion on which of these two ways of writing it you feel would be most appropriate. The approach from [1] is definitely more formal and descriptive of how it is measured in practice. On the other hand, we also worry about bogging the readers down in notation like the notion of paths through the MDP, which could distract readers from our main contributions.
>
> [1] Michael Kearns and Satinder Singh. Near-optimal reinforcement learning in polynomial time. *Machine learning,* 49(2):209–232, 2002.

---

### Decision · Program_Chairs · 2024-09-25

**Decision:**

Accept (poster)

**Comment:**

The paper studies the trade-off between increasing context length in reinforcement learning (RL) models and the resulting increase in mixing time, which affects how quickly the model can be evaluated and trained. The paper presents theoretical insights and empirical evidence showing that although larger context lengths can improve policy performance, they also make reliable evaluation more challenging, especially in large-scale and complex environments.

This paper will provide refreshing insights to some and technical confirmation of a hunch to others. Overall, the reviewers appreciated both the theoretical and the empirical results and their significance. There were concerns whether high mixing times are a significant issue in real-world applications. I find the argument authors gave convincing: “We believe the settings of greatest relevance to our work are those related to continual or multi-task environments where agents are evaluated as generalists over a number of skills rather than just solving a single narrow task. As such, we believe that focus on the difficulties presented by high mixing times is timely in the age of foundation models.” I suggest the authors incorporate these arguments into their work. I also suggest the authors to incorporate the feedback from the reviewers such as adding a proof sketch and visualizing attention maps of transformer.